# Progress in Pharmacokinetics, Pharmacological Effects, and Molecular Mechanisms of Swertiamarin: A Comprehensive Review

**DOI:** 10.3390/cells14151173

**Published:** 2025-07-30

**Authors:** Hao-Xin Yang, Ying-Yue Hu, Rui Liang, Hong Zheng, Xuan Zhang

**Affiliations:** 1Yunnan Key Laboratory of Pharmacology for Natural Products, School of Pharmaceutical Sciences, Kunming Medical University, Kunming 650500, China; 20230114@kmmu.edu.cn (H.-X.Y.); 20230145@kmmu.edu.cn (Y.-Y.H.); 20240122@kmmu.edu.cn (R.L.); 2Department of Laboratory Animal Science, Kunming Medical University, Kunming 650500, China; 3Yunnan College of Modern Biomedical Industry, Kunming Medical University, Kunming 650500, China

**Keywords:** swertiamarin, secoiridoid glycoside, pharmacokinetic properties, multiple pharmacological effects, molecular mechanisms

## Abstract

Swertiamarin (SW), a natural iridoid glycoside primarily isolated from the genus *Swertia*, *Gentianaceae* family, has been extensively utilized in traditional medicine systems, including Ayurveda, Traditional Chinese Medicine, and Tibetan medicine, for treating fever, diabetes, liver disorders, and inflammatory conditions. Pharmacokinetic studies reveal that SW exhibits rapid absorption but demonstrates low oral bioavailability due to the first-pass effect. Pharmacological studies have demonstrated that SW possesses a wide range of pharmacological activities, including antioxidant, anti-inflammatory, anti-tumor, anti-diabetic, and neuroprotective activities. Our analysis demonstrates that SW exerts remarkable therapeutic potential across multiple pathological conditions through coordinated modulation of key signaling cascades, including Nrf2/HO-1, NF-κB, MAPK, PI3K/Akt, and PPAR pathways. This comprehensive review systematically consolidates current knowledge on SW’s pharmacokinetic characteristics, toxicity, diverse biological activities, and underlying molecular mechanisms based on extensive preclinical evidence, establishing a scientific foundation for future drug development strategies and potential clinical applications of the potential natural lead compound.

## 1. Introduction

Swertiamarin (SW) is a natural iridoid glycoside primarily isolated from the genus *Swertia*, *Gentianaceae* family [1]. As a medicinal plant of natural origin, extracts from *Gentianaceae* possess various pharmacological activities, including anti-inflammatory and antioxidant, hepatoprotective, and anti-diabetic effects [2,3,4,5,6,7,8,9]. One of its main active components, swertiamarin, also plays a significant therapeutic role in treating diabetes mellitus, cardiovascular diseases, and neurological disorders. In Table 1, we summarize the sources, medicinal parts, and pharmacological effects of plant extracts containing SW. Furthermore, research on its pharmacokinetic properties and toxicological safety continues to advance. Previous reviews have highlighted its anti-diabetic and anti-hyperlipidemic effects [10] and effects on metabolic diseases [11,12,13]. However, previous studies have lacked systematic integration of the pharmacokinetics, pharmacological effects, and molecular mechanisms of SW. In this paper, we will focus on a systematic review of the pharmacokinetic properties, pharmacological effects, and molecular mechanisms associated with swertiamarin, providing a reference for its clinical application and development. Figure 1 shows the chemical structure of SW and the structures of its metabolites and analogues.

**Table 1 cells-14-01173-t001:** Plant sources, medicinal parts, and pharmacological effects of plant extracts containing SW.

Plant Sources	Part	Pharmacological Effects	Refs.
*Swertia mussotii* Franch	Whole plants	Hepatoprotective effects	[17,18]
*Swertia pseudochinensis* Hara	Whole plants	Anti-cholestasis effects	[19]
*Enicostemmalittorale* Blume	Whole plants	Anti-diabetic effects	[20]
*Lomatogonium rotatum*	Whole plants	Anti-diabetic effects	[21]
*Enicostemma axillare*	Whole plants	Antioxidant effects	[22]
*Enicostemmalittorale* Blume	Whole plants	Anti-hyperlipidemic effect	[23]
*Enicostemma axillare*	Whole plants	Antinociceptive activity	[24]
*Gentiana straminea*	Roots	Antioxidant effects	[25]
*Gentiana kurroo*	Roots	Anti-cancer effects	[26]
*Swertia japonica* Makino	Whole plants	Stimulate the gastric emptying effect	[27]
*Centaurium pulchellum*	Aerial parts and roots	Antibacterial and antifungal effects	[28]

## 2. Pharmacokinetic Properties

The analysis of pharmacokinetic data showed that SW was rapidly absorbed into the circulatory system after oral administration of SW (20 mg/kg) in SD rats, reaching peak concentrations in approximately 0.95 h, and had a short half-life. The terminal half-fife (t_1/2z_ (h) is 1.104 ± 0.22, V_z/F_ (L/kg) is 9.637 ± 4.322, the apparent terminal clearance (CL_z/F_) (L/h/kg) is 5.638 ± 2.151, the area under the curve (AUC) (μg/L h) is 3593.7 ± 985.4, MRT_0–∞_ (h) is 1.929 ± 0.364, T_max_ (h) is 0.945 ± 0.136, and the maximum plasma concentration (C_max_) (μg/L) is 1920.1 ± 947.0 [29]. LC-MS/MS determination revealed that the absolute oral bioavailability of SW (25 mg/kg) in SD rats was low at 10.3%, which is attributed to its poor permeability through the intestinal epithelial membrane and first-pass effect in the liver. Following a single oral dose of SW (50 mg/kg) to rats, it was rapidly distributed throughout the tissues and peaked at 1 h. Its concentrations are relatively high in the kidney and liver, suggesting that SW is rapidly metabolized in the liver and excreted by the kidneys, while its levels in the brain are notably low due to its high hydrophilicity, which hinders its ability to cross the blood–brain barrier [30,31].

The pharmacokinetic (PK) parameters of SW are also affected by various combinations of drug administration. Studies on the pharmacokinetics of the three SW (20 mg/kg) administration types in rats showed that Qing Ye Dan tablets (QYDTs) and co-administration of SW and oleanolic acid significantly decreased the peak plasma concentration and AUC, as well as increased CL and V_z/F_ compared with oral administration of SW alone. These data indicate that oleanolic acid or another component of QYDTs might reduce the oral bioavailability of SW in rats [32]. Furthermore, analysis of the PK parameters of four components (gentiopicroside, geniposide, baicalin, and swertiamarin) in rat plasma after oral administration of the Chinese herbal prescription Long-Dan-Xie-Gan-Tang (LDXGT) (10 g/kg, p.o.) showed that SW exhibited the lowest level of AUC (2.5 ± 0.1 min µg/mL) and fell below the lower limit of quantification (LLOQ) [31]. Additionally, pathological conditions and the compatibility effects of other ingredients in Chinese herbal formulations could also affect the pharmacokinetics of SW. There were significant differences in PK parameters between rats with complete Freund’s adjuvant (CFA)-induced arthritis and normal rats orally administered Huo Luo Xiao Ling Dan (HLXLD); the AUC_0–t_, area under the curve (AUC_0–∞_), and C_max_ of the three secoiridoid glycosides (gentiopicroside, swertiamarin, sweroside) decreased to varying degrees after oral administration of HLXLD as compared with the normal control group, while the clearance of these glycosides increased significantly. The results indicate that the three components were poorly absorbed and rapidly metabolized in the RA model. Meanwhile, the three secoiridoid glycosides in the HLXLD group had a longer T_max_ compared with a single-herb extract of GME, suggesting slower absorption and distribution processes in the HLXLD group [33].

After oral administration of three different doses of Swertia cincta solution (0.5 g/kg, 1 g/kg, 2 g/kg), SW showed a dose-dependent nonlinear pharmacokinetic profile in rats, with the increase in AUC and C_max_ being lower than that of the dose, which was related to saturation of absorption or an enhancement in the first-pass effect. And the absorption of SW was slower, but its elimination was faster compared with other active ingredients (sweroside, gentiopicroside) in Swertia cincta extract [34]. Studies have shown that there are significant gender differences in the pharmacokinetic properties of SW in rats. The C_max_, AUC_0-t_, AUC_0–∞_, and t_1/2_ values were significantly higher in female rats than in male rats after oral administration of SW (1.9 mg/kg). However, the elimination rate (CL) was significantly lower in females than in males, which indicated that SW is absorbed to a greater extent and eliminated more slowly in females compared with males [35].

After a single oral administration of 4 mL/kg of *Swertia pseudochinensis* extract to rats, the plasma concentration–time curve of SW exhibited a double-peak phenomenon, which was attributed to factors such as enterohepatic recirculation and the physicochemical properties of the compound. Excretion studies indicated that six analytes, including SW, were predominantly excreted in urine. However, the cumulative amounts of intact SW and the other five analytes excreted via bile, urine, and feces accounted for only a minor fraction of the administered dose, suggesting that these compounds were primarily eliminated as metabolites [36]. Metabolic analyses in Wistar rats after oral administration of SW showed that SW (**1**, Figure 2) was initially hydrolyzed by β-glycosidase in the intestine, and due to the instability of aglycone, it spontaneously reacted with ammonia to form gentianine (**2**, Figure 2). Gentianine was further converted to the nitrogen-containing metabolite gentiandiol (**3**, Figure 2) in the liver and ultimately excreted through urine; its proposed metabolic pathway is shown in Figure 2 [14]. The endophyte *P. brasilianum* can biotransform SW into a range of active metabolites, among which the metabolite M01 (gentianine) has anti-diabetic potential due to its nitrogen-substituted structure [37,38]. In addition, the aglycone of SW hydrolyzed by bacterial β-glucosidase is readily converted to erythrocentaurin (ECR). Some ECR is further reduced to 3,4-dihydro-5-(hydroxymethyl) isochroman-1-one (HMIO) by both liver and intestinal bacteria. Therefore, SW can be metabolized to dihydroisocoumarin and alkaloid compounds in vivo to exert pharmacological effects [39,40]. Pharmacokinetic studies of rat plasma based on LC-MS/MS showed that SW was absorbed and eliminated relatively quickly after oral administration. The oral bioavailability of SW in SD rats decreased with increasing dose (8.0% at 50 mg/kg, 6.7% at 100 mg/kg, and 6.2% at 150 mg/kg). Further studies showed that a total of six metabolites were isolated and identified in serum, urine, bile, and feces by using UPLC-Q/TOF-MS/MS technology. SW and all metabolites were classified into cleavage products of de-hydroxylation of aglycones, the isomerization product after dehydration of aglycones, and the aglycon heterocyclic product, which first formed aglycones before the next metabolism [41]. Another study showed that 49 metabolites altogether, including the archetype compound of SW, were found in SD male rats following oral administration. Furthermore, the in vivo biotransformation process mainly involved phase I reactions, such as reduction, dehydration, hydroxylation, and phase II reactions, such as sulphonation and N-acetylcysteine (NAC) formation. Drug metabolic clusters (DMCs) centered on gentianine or other compounds were also detected [42].

## 3. Pharmacological Effects

### 3.1. Hepatoprotective Effects

An earlier study showed that, in a rat model of D-galactosamine-induced acute liver injury, oral administration of SW (100 and 200 mg/kg) for 8 days exerted hepatoprotective effects by attenuating morphological changes, such as liver tissue necrosis and bile duct proliferation, restoring biochemical indexes towards normal levels, alleviating D-GalN-caused hepatotoxicity, enhancing the levels of antioxidant enzymes catalase (CAT), superoxide dismutase (SOD), and glutathione (GSH), and decreasing the levels of thiobarbituric acid reactive substances (TBARSs) [43]. SW obviously attenuated the inflammatory response of CCl_4_-induced liver injury in rats, as evidenced by a significant reduction in myeloperoxidase (MPO) activity, inducible nitric oxide synthase (iNOS) activity, nitric oxide (NO) levels, and the content of pro-inflammatory factors interleukin 1beta (IL-1β), tumor necrosis factor-alpha (TNF-α), and IL-6 in the liver after SW treatment, as well as remarkably repressing hepatic oxidative stress, inducing the expression of hepatic detoxification enzymes CYPs, hepatic efflux transporter protein bile salt export pump (Bsep), multidrug resistance-associated protein 2 (Mrp2), Mrp3, Mrp40, and PDZ domain-containing protein 1(PDZK1). In addition, co-treatment of SW with CCl_4_ notably upregulated the expression of nuclear factor erythroid 2-related factor 2 (Nrf2), heme oxygenase-1 (HO-1), and NAD(P)H quinone dehydrogenase (NQO1). These results suggested that the protective effect of SW against CCl_4_-induced liver injury might be linked with the activation of the Nrf2/HO-1 pathway [44]. In CCl_4_-induced liver injury mice, SW supplementation exhibited a lower liver weight and liver index, mitigating cell degeneration, inflammatory cell infiltration, and collagenous fiber deposition. Mice treated with SW significantly reversed the elevation in alanine aminotransferase (ALT), aspartate aminotransferase (AST), malondialdehyde (MDA), and hydroxyproline (Hyp) levels induced by CCl_4_. Furthermore, multi-omics analysis revealed that SW could ameliorate CCl_4_-induced liver toxicity by regulating the gut microbiota and its metabolites [45]. SW also attenuated carbon tetrachloride-induced apoptosis in rat hepatocytes and hepatic stellate cell activation via increasing B-cell lymphoma 2 (Bcl-2) levels and inhibiting the expression and activation of caspase-3. This anti-apoptotic effect was related to the regulation of the PI3K/Akt (phosphoinositide 3-kinase/protein kinase B) signaling pathway [46]. In addition, SW ameliorated nicotine-induced damage in SD rats by regulating oxidative stress [47].

SW is an iridoid glycoside compound that is unstable to heat and prone to decomposition and transformation [48]. The heat-transformed products (HTPs) of SW are mainly responsible for its hepatoprotective effects. Sweritranslactone D, a novel secoiridoid dimer isolated from the HTPs, exhibited more potent activity in alleviating liver injury than N-acetyl-L-cysteine in the L-O2 AP-induced cell model [15]. Another study confirmed that both SW and its metabolite HTPs protect against acetaminophen (APAP)-induced hepatotoxicity by inhibiting inflammatory responses and oxidative stress while regulating the nuclear factor erythroid 2-related factor/nuclear factor kappa-light-chain-enhancer of activated B cells (Nrf-2/NF-κB) signaling pathway. In vitro, the compounds reduced apoptosis in APAP-exposed L-O2 hepatocytes. In vivo, they attenuated histological liver damage and decreased injury biomarkers in mice [49].

Liver fibrosis is an abnormal wound-healing response caused by many types of acute and chronic liver injuries, which may further progress to cirrhosis, liver failure, and liver cancer [50]. In vitro, SW significantly inhibited angiotensin II (Ang II)-induced primary rat hepatic stellate cells (HSCs) proliferation and activation, downregulating transforming growth factor-beta 1 (TGF-β1) mRNA level. In vivo, SW significantly improved dimethylnitrosamine (DMN)-induced liver function injury and ameliorated liver fibrosis by reducing collagen deposition, inhibiting α-smooth muscle actin (α-SMA) expression, and reducing the Hyp content in rat livers. This antihepatic fibrosis effect is mainly mediated by modulation of the renin–angiotensin system (RAS), including the downregulation of Ang II and angiotensin II type 1 receptor (AT1R) expression and inhibition of extracellular signal-regulated kinase (ERK) and c-jun phosphorylation [51].

Nonalcoholic fatty liver disease (NAFLD) is one of the most common chronic liver diseases, with a worldwide prevalence of up to 25% [52]. In fructose-fed mice, the administration of swertiamarin (25, 50, and 100 mg/kg) significantly decreased serum triglyceride (TG), glucose (GLU), uric acid (UA), AST, and ALT levels. SW ameliorated fructose-induced liver injury by inhibiting hepatic ballooning degeneration and fatty deposition and reducing liver tissue TG levels. Further studies have shown that SW could attenuate NAFLD and metabolic alterations in fructose-fed mice by inhibiting hepatic inflammatory responses and xanthine oxidase (XO) activity, enhancing the antioxidant defense system, reducing hepatic steatosis, and downregulating the expression of lipogenesis-controlling factors, including sterol regulatory element-binding protein 1 (SREBP-1), fatty acid synthase (FAS), and acetyl-CoA carboxylase 1 (ACC1) [53]. In oleic acid (OA)-induced insulin resistance HepG2 cells, SW attenuated hepatic glycemic load, lipid accumulation, reactive oxygen species (ROS), and insulin resistance by targeting AMP-activated protein kinase (AMPK) and peroxisome proliferator-activated receptor α (PPAR-α) and modulating fat metabolic enzymes, such as FAS, ACC1, phosphoenolpyruvate carboxykinase (PEPCK), and the expression levels of insulin signal proteins, such as ser-307 insulin receptor substrate 1 (IRS1), insulin receptor β-subunit (IR-β), phosphatidylinositol 3-kinase (PI(3)K), and phosphorylated AKT (p-AKT). Swertiamarin provides a potential source of drugs for the treatment of hepatic steatosis [54].

Cholestasis is a pathological state caused by impaired bile secretion and excretion. Prolonged cholestasis can also lead to liver failure and cirrhosis, increasing the risk of liver fibrosis and liver cancer [55,56]. Metabolomics analyses of the α-naphthylisothiocyanate (ANIT)-induced cholestasis rat model revealed that SW altered multiple metabolomic pathways related to cholestasis, including primary bile acid synthesis, retinol metabolism, glycerophospholipid metabolism, and sphingolipid metabolism. SW also reversed changes in multiple protein targets, such as CYP7A1, BSEP, Na^+^-taurocholate cotransporting polypeptide (NTCP), small heterodimer partner (SHP), and multidrug resistance-associated protein 2 (MRP2), mainly focusing on the farnesoid X receptor (FXR). These results suggested that SW can mitigate ANIT-induced cholestasis by activating FXR and bile acid excretion pathways [19]. In the ANIT-induced rat model of cholestatic hepatitis, the total iridoid and xanthone extract from *Swertia mussotii* Franch (TIXS) promoted bile secretion and lowered the activities of the serum enzymes ALT, AST, and alkaline phosphatase (ALP), as well as the levels of the bilirubins TBIL, DBIL, and UCBIL, thus exerting a therapeutic effect on cholestatic hepatitis. Swertiamarin and swertianolin are the active constituents of TIXS [18]. Bile duct ligation (BDL) rats administered swertiamarin showed low levels of ALT, AST, TNF-α, IL-1, and IL-6. SW reduced the toxic bile salt concentrations in the serum of cholestatic rats, including chenodeoxycholic acid (CDCA) and deoxycholic acid (DCA) [17]. These studies have elucidated the protective role of SW in cholestasis. Figure 3 presents the important mechanisms and targets where SW exerts hepatoprotective effects.

### 3.2. Anti-Diabetic Effects

Diabetes is a highly prevalent disease that was originally classified into three main categories: type 1 diabetes, type 2 diabetes, gestational diabetes mellitus, and specific types of diabetes due to other causes. It is estimated that type 2 diabetes accounts for 90–95% of all cases of diabetes [57]. Previous studies have demonstrated that the plant extract of *Enicostemma littorale* possesses hypoglycemic effects, which are mainly attributed to its contents of the active ingredient SW. SW and its derivatives can exert their anti-diabetic effects through the modulation of multiple targets, including adiponectin, peroxisome proliferator-activated receptor γ (PPARγ), glucose transporter 4 (GLUT4), GLUT2, Phosphoenolpyruvate carboxykinase (PEPCK), Glucokinase (GK), HMG-CoA Reductase (HMGR), Acetyl-CoA Carboxylase, 5-hydroxytryptamine 2 receptor (5-HT2) [58], and other molecular targets [10]. In STZ-induced T2DM Wistar rats, the combination of swertiamarin and quercetin (CSQ) exerted synergistic therapeutic effects. The CSQ significantly ameliorated metabolic dysfunction by reducing fasting blood glucose, TGs, total cholesterol (TC), and low-density lipoprotein cholesterol (LDL) while restoring insulin secretion and normalizing carbohydrate-metabolizing enzymes. The CSQ also enhanced antioxidant capacity by increasing the activities of serum SOD, CAT, GSH, and glutathione peroxidase (GPx). Meanwhile, it decreased lipid peroxidation, and promoted the regeneration of pancreatic islets and insulin secretion. Collectively, these multi-targeted actions establish the CSQ as a promising combinatorial strategy for T2DM management [59]. In nicotinamide–streptozotocin (NA-STZ)-induced diabetic rats, SM inhibited HMG-CoA reductase activity in dyslipidemic conditions, improved insulin sensitivity, and modulated carbohydrate and fat metabolism by modulating PPAR-γ and other transcription factors. SW could be an effective therapeutic agent for TIIDM [60]. SW increased glucose consumption by activating the PI3K/AKT pathway in HepG2 cells, thereby ameliorating dexamethasone-induced insulin resistance and exerting a hypoglycemic effect [61]. SW treatment for 28 days significantly reduced serum urea and creatinine levels, restored glucose metabolizing enzymes, promoted islet regeneration, and ameliorated diabetes-induced pancreatic, cardiac, hepatic, and renal lesions in STZ-induced diabetic rats [62].

As swertiamarin has a very short plasma half-life, the anti-diabetic effect of swertiamarin is related to its active metabolite, gentianine. Treatment with gentianine induced adipogenesis and differentiation by upregulating the gene expression of PPAR-γ, GLUT-4, and adiponectin. Adiponectin mRNA expression was also significantly increased by swertiamarin treatment, thereby enhancing insulin sensitivity and exerting anti-diabetic effects [37]. In streptozotocin-induced diabetic rats fed a high-fat diet, treatment with SW significantly decreased the level of glucose to 142.8 mg/dl and also reduced the levels of serum TG and cholesterol. Treatment with SW significantly ameliorated renal function and renal fibrosis in rats by preventing hypertrophy in the kidneys and decreasing the levels of the kidney index, blood creatinine, and blood uric acid. SW mitigated the levels of advanced glycation end products (AGEs) in the serum and kidneys of diabetic rats. Additionally, it remarkably reduced MDA levels in the serum and kidneys of rats, indicating that it effectively alleviates oxidative stress induced by hyperglycemia and AGEs. Treatment with SW could also reduce the levels of advanced glycation end products (RAGE), p38 mitogen-activated protein kinase (p38 MAPK), and nicotinamide adenine dinucleotide phosphate (NADPH), blocking the binding of AGEs with RAGE [63]. *Lomatogonium rotatum* (LR) is an important herb containing the active ingredient SW, which prevents high-fat, high-glucose diet and STZ-induced diabetes in rats by altering the levels of serum metabolites, such as vitamin B6 and mevalonate-5P, and promoting the release of insulin and GLP-1 [21].

Diabetes can cause complications in multiple systems and organs, and the main risks include retinopathy, diabetic nephropathy (DN), neuropathy, and cardiovascular disease [64]. Numerous studies have indicated that SW also has an excellent curative effect on diabetic complications. SW treatment (50 mg/kg, i.p.) reduced serum triglyceride, cholesterol, and low-density lipoprotein (LDL) levels in diabetic animals and significantly increased the insulin sensitivity index, thereby improving the dyslipidemic condition developed as a result of insulin resistance [65]. SW exerted anti-glycosylation effects by inhibiting the formation of fructose-induced AGEs and the binding of AGEs with RAGE. SW inhibited the expression of inflammatory factors IL-6, TNF-α, and IL-1β in methylglyoxal (MG)-treated NRK-52E cells and reduced oxidative stress in renal cells by upregulating the levels of Nrf-2 and HO-1, which reduced TGF-β activation and improved the epithelial–mesenchymal transition (EMT) in MG-induced NRK-52E. The inhibition of the RAGE/MAPK/TGF-β pathway is one of the mechanisms by which SW prevents DN in MG-induced kidney cells [66]. Oral administration of swertiamarin (50 mg/kg) significantly reduced serum urea and creatinine levels and ameliorated glomerular injury and other parameters associated with the development of DN in type 1 diabetic rats [20]. Diabetic peripheral neuropathy (DPN) is one of the common complications of diabetes mellitus, with a reported incidence of 13–26% [67,68]. In a streptozotocin (STZ)-induced rat model of DPN, SW maintained inflammatory factor homeostasis by inhibiting the signaling pathway of the NADPH oxidase (NOXS)/ROS/NOD-like receptor protein 3 (NLRP3) inflammatory cascade response to reduce hyperalgesia and effectively protected nerves for the treatment of DPN [69]. In Zucker fa/fa rats, treatment with SW significantly reduced serum glucose, serum cholesterol, TG, and non-esterified fatty acid (NEFA) levels and lowered serum matrix metalloproteinase-9 (MMP-9), MMP-3, and serum urea levels; it also had a significant protective effect against diabetes-induced cardiovascular complications, such as atherosclerosis and nephropathy [70]. Figure 4 summarizes the multiple mechanisms by which SW exerts anti-diabetic effects.

### 3.3. Neuroprotective Effects

The incidence of neurodegenerative diseases is increasing with the aging population. Recently, SW demonstrated neuroprotective effects in vivo and in vitro. In lipopolysaccharide (LPS)-induced activation, SW (10–100 μg/mL) treatment significantly reduced the levels of pro-inflammatory cytokines (IL-1β, IL-6, and TNF-α). In a rotenone-induced mouse model of PD, SW inhibited microglial and astrocyte activation in the substantia nigra (SN) and reduced alpha-synuclein (α-syn) overexpression in the striatum and SN. SW also alleviated rotenone-induced motor impairment. Furthermore, SW increased tyrosine hydroxylase (TH) immunoreactivity in the striatum and the number of TH+ neurons in the SN [71,72]. In the Caenorhabditis elegans model, SW enhanced neurotransmission by modulating nicotinic acetylcholine receptor (nAChR) and acetylcholinesterase (AChE) activities, attenuated ROS levels, and increased antioxidant enzymes levels by upregulating sod-3 and gst-4 levels, thereby ameliorating cholinergic dysfunction and exerting a protective effect against neurodegeneration [73]. In the 3-NP-induced Huntington’s disease (HD) rat model, administration of SW attenuated 3-nitropropionic acid (3-NP)-induced memory deficits, behavioral abnormalities, and biochemical parameters alterations, partially reduced ChE levels in the cerebral cortex and hippocampus of HD rats, and significantly ameliorated mitochondrial complexes enzyme activities and oxidative damage in various brain regions [74]. Therefore, it may be an effective neuroprotective agent for HD and other degenerative diseases.

Moreover, SW has significant anti-inflammatory activity, which suggests its potential protective role in neuro-inflammation. In BV-2 cells treated with LPS, SW (10, 25, 50 μg/mL) differentially inhibited the inflammatory cytokine secretion of IL-1β, IL-6, IL-18, and TNF-α in BV-2 cells. Proteomics analysis results suggested that the potential bio-processes regulated by SW were mainly involved in the cellular response to carbon monoxide, strand displacement, palmitoleoyltransferase activity, D2 dopamine receptor binding, and RNA polymerase II transcription cofactor activity [75]. SW also inhibited apoptosis and ROS production via the toll-like receptor 4 (TLR4)/poly (ADP-ribose) polymerase 1(PARP1)/NF-κB pathway, thereby protecting human neuronal SH-SY5Y cells from oxygen–glucose deprivation/re-oxygenation (OGDR)-stimulated damage [76]. In α-syn-expressing worms, the expression of fat-5 and fat-7 lipid levels were significantly reduced after SW treatment, thereby reducing α-syn deposition. At the same time, it increased mitochondrial viability, decreased the levels of ROS, reduced apoptosis, significantly elevated dopamine (DA) levels, and modulated dopamine-dependent behavior in NI 5901 worms. In BZ555 worms induced by the neurotoxin 6-OHDA, SW remarkably protected against neurotoxin-induced dopaminergic neuronal degeneration. The neuroprotective effect of SW can upregulate the expression of SKN-1 and GST-4 through the MAPK pathway [77]. SW also exerts antidepressant effects by modulating 5-HT2 receptors [58].

Stroke is one of the cerebrovascular diseases with high global morbidity and mortality. Thrombolysis is an effective treatment for ischemic stroke, while cerebral ischemia–reperfusion injury(CIRI) is a major complication of treatment [78]. In the mouse middle cerebral artery occlusion (MCAO) model, at 24 h after reperfusion, pre-treatment with SW (25, 100, or 400 mg/kg) dose-dependently markedly reduced infarct volume, reversed the change in pathology produced by CIRI to some extent, and significantly inhibited I/R-induced neuronal apoptosis by increasing the levels of the anti-apoptotic protein Bcl-2 and decreasing Bcl-2-associated X protein (Bax) levels. SW also protected against I/R-induced oxidative stress damage by reducing MDA levels and upregulating antioxidant enzyme (GSH-PX, SOD, and CAT) activities in vivo. In addition, SW treatment decreased cell death and intracellular ROS levels induced by oxygen–glucose deprivation/reperfusion (OGD/R) in primary cultured hippocampal neurons. It promoted Nrf2 nuclear translocation from the Keap1-Nrf2 complex and increased the expression of Nrf2, HO-1, and NQO1 in the mouse MACO model and the primary hippocampal neuronal cell OGD/R model. These results suggest that the neuroprotective effect of SW against I/R injury is dependent on the activation of Nrf2 pathway-mediated anti-oxidative stress, which may be a promising protective agent against ischemic brain damage [79].

In pilocarpine (PILO)-treated mice, SW pre-treatment significantly delayed the first convulsion and reduced the incidence of persistent status epilepticus and mortality, as well as reduced neuronal loss and neuronal damage in hippocampal cornu ammonis 1 (CA1) and CA3 regions and inhibited astrocyte activation and inflammatory responses [80]. SW also stimulated gamma-aminobutyric acid (GABA) receptors and significantly inhibited pentylenetetrazol (PTZ)-induced convulsion seizures in albino mice [81], suggesting its potential to prevent and attenuate epileptic seizures. The multiple mechanisms of SW’s neuroprotective effects are shown in Figure 5.

### 3.4. Regulation of Lipid Metabolism

Disorders of lipid metabolism are one of the risk factors for many cardiovascular diseases, such as hyperlipidemia, atherosclerosis, and coronary heart disease [82,83]. In high-cholesterol-fed rats, oral administration of SW (50 and 75 mg/kg) significantly reduced serum levels of TC, TG, LDL, and VLDL while inhibiting hepatic 3-hydroxy-3-methylglutaryl coenzyme A (HMG-CoA) reductase activity. SW also enhanced fecal excretion of bile acids and total sterols via upregulation of 17α-hydroxylase activity [23]. Separately, in poloxamer-407-induced hyperlipidemic rats, a single oral dose of SW (50 mg/kg) significantly decreased the LDL/HDL cholesterol ratio at both 15 h and 24 h post-dose [84]. Another study demonstrated that SW could also decrease fat storage in vivo. Knockdown of kat-1 inhibited the lipid-lowering effect of SW, whereas SW treatment significantly increased kat-1 expression. These results indicated that SW regulates lipid homeostasis and exerts lipid-lowering effects dependent on kat-1 [85]. These results demonstrated the potential lipid-lowering and anti-atherosclerosis activities of SW. Aqueous root extracts containing SW inhibited platelet-derived growth factor (PDGF)-induced activation of Extracellular signal-regulated kinase 1/2 (ERK1/2) and proliferation of rat aortic smooth muscle cells (RASMCs), suggesting their potential as new drug candidates for the treatment of atherosclerosis [25].

Obesity is one of the global public health problems, and it is closely related to diabetes and cardiovascular disease [86]. In high-fat diet (HFD)-induced obese C57BL/6 mice, SW attenuated HFD-induced weight gain, glucose intolerance, and obesity-related insulin resistance by enhancing insulin signaling. Treatment with SW notably increased lipolysis and reduced adipocyte hypertrophy and macrophage infiltration in the epididymal white adipose tissue (eWAT). SW supplementation decreased the expression of inflammatory cytokines (TNF-α and IL-1β) and chemokines (Ccl2 and Ccl5) in LPS-stimulated macrophages and in HFD-fed mice, which also further indicated that SW improved obesity-associated chronic inflammation and hepatic steatosis by suppressing activation of the p38 MAPK and NF-κB pathways [87]. Another study showed that SW treatment for 10 weeks significantly reduced body weight and enhanced energy expenditure by promoting brown adipose tissue (BAT) activation and fat browning in diet-induced obese mice, as well as improved the irregular fiber structure and excess lipid deposition and oxidative metabolism by increasing fatty acid oxidation in skeletal muscle [88]. The above studies highlight the potential clinical utility of SW in the prevention of obesity and related metabolic disorders. In Figure 6, we summarize the regulatory role of SW in lipid metabolism.

### 3.5. Anti-Inflammatory and Immunomodulation

In SRBC-immunized mice, SW treatment (2, 5, and 10 mg/kg) significantly increased antibody titer, plaque-forming cells, and immune organ weights and attenuated delayed-type hypersensitivity (DTH). In vitro, SW significantly inhibited free radical release in PHA-stimulated neutrophils and reduced the expression of pro-inflammatory factors in LPS-induced macrophages. These findings suggest that SW exerts its anti-inflammatory effects by modulating both humoral and cell-mediated immune responses, suppressing pro-inflammatory mediators (Th1-type cytokines), and promoting anti-inflammatory mediators (Th2-type cytokines) [89]. As a potential natural AKT inhibitor, SW could directly target the structural domain of AKT-PH to inhibit AKT activation and downregulate the phosphorylation level of AKT, which further inhibits the production of downstream inflammatory molecules, thus presenting significant anti-inflammatory activity in cells and animal models of acute lung injury [90].

In the *Plasmodium berghei*-infected Swiss albino mouse model, oral administration of SW (200 mg/kg) significantly reduced parasite load and mitigated the severity of malaria infection. SW also enhanced the production of immunomodulatory mediators in vivo and effectively modulated the levels of interferon-gamma (IFN-γ), IL-10, and TNF-α to exert antimalarial protection, contributing to its antimalarial protection [91]. Another study showed that rutin and swertiamarin displayed significant synergistic antimalarial activity in vivo and in vitro. When combined in a 1:1 ratio, they reduced the IC_50_ value of Plasmodium falciparum. In *Plasmodium berghei*-infected mice, they achieved 79.95% chemosuppression with a survival time comparable to that of mice treated with chloroquine phosphate [92]. Therefore, SW may serve as an immunomodulatory agent for the combination treatment of malaria infection.

Chlorogenic acid (CA) and SW, as key compounds in Honeysuckle Extract Preparation (HEP), synergistically inhibited pro-inflammatory cytokines IL-1 and IL-6 and prostaglandin E2 (PGE2) production by targeting the PI3 K-AKT and p38 MAPK pathways, thereby alleviating LPS-induced fever [93]. SW could also exert antiedematogenic effects through anti-inflammatory and free radical scavenging in rat paw edema models [22]. Figure 7 lists the multiple mechanisms and targets by which SW exerts anti-inflammatory and immunomodulatory effects.

### 3.6. Anti-Tumor Effects

Studies have demonstrated that SW possesses anti-tumor activity. SW significantly inhibited the viability and invasion of HepG2 liver cancer cells and promoted apoptosis. In vivo, SW also notably suppressed the growth of SK-Hep-1 cells xenografted in nude mice. Bioinformatics analysis has identified PI3K, transcription factor AP-1 (JUN), and signal transducer and activator of transcription 3 (STAT3) as predictive targets through which SW exerts its anti-tumor effects in hepatocellular carcinoma (HCC) [94]. Further investigations have shown that SW inhibited HepG2 and Huh7 liver cancer cell proliferation, migration, and invasion by downregulating frequently rearranged in advanced T-cell lymphomas 1 (FRAT1) and partially suppressing the FRAT1/Wnt/β-catenin signaling axis [95]. These data demonstrated the therapeutic potential of SW for HCC. SW also showed significant antiproliferative activity on the HEp-2 and HT-29 cancer cell lines [81]. *Gentiana kurroo* root extract, with SW as its major active component, exerted multiple anti-cancer effects in the human pancreatic cancer cell line Miapaca-2. These included G0/G1 cell cycle arrest, protection of DNA from oxidative damage, and induction of apoptosis via disruption of the mitochondrial membrane potential (ΔΨm) [26]. Swertiamarin B, a novel secoiridoid compound isolated from *Swertia mussotii* Franch, effectively inhibited the proliferation of gastric cancer cells [16]. In studies of human peripheral blood mononuclear cells (PBMCs), the expression levels of PARP1 and Bax/Bcl-2 were elevated after SW treatment, suggesting that SW can activate apoptosis and necroptosis to eliminate malignant cells [96]. SW bonded strongly to both human serum albumin (HSA) and α-1 acid glycoprotein (AGP), inhibited proliferation, and induced apoptosis of the neuroblastoma (SK-N-AS) cell line [97]. In Figure 8, we summarize the broad spectrum of anti-tumor activity of SW in several types of cancer cells.

### 3.7. Bone and Joint Protective Effects

Rheumatoid arthritis (RA) is a chronic autoimmune disease characterized by synovial inflammation [98]. SW also has a significant therapeutic effect on RA. In IL-1β-induced AA-FLS cells (adjuvant-induced arthritic rat fibroblast-like synoviocytes), SW treatment significantly inhibited proliferation and NO production, increased mRNA and protein expression levels of apoptosis mediator caspase-3 in a dose-dependent manner, decreased the levels of the inflammatory mediator iNOS, cyclooxygenase-2 (COX-2), PGE2, TNF-α, and IL-6, and also modulated the balance of MMPs/Tissue inhibitors of metalloproteinases (TIMPs). The treatment significantly suppressed the release of osteoclastogenic mediator receptor activator of NF-κB ligand (RANKL) and p38 MAPKα. These results indicated that SW could alleviate IL-1β-induced AA-FLS from joint destruction by exerting anti-inflammatory, pro-apoptotic, and anti-osteoclastogenic effects [99]. Bone erosion is a crucial and destructive pathological feature in the disease process of RA. It signifies structural destruction of the joints [100]. In vivo, treatment with SW significantly reversed pathological alterations caused by Freund’s complete adjuvant (FCA) stimulation in animals, including calcium and phosphorus, bone collagen, tartrate-resistant acid phosphatase (TRAP), acid phosphatase (ACP), and alkaline phosphatase (ALP) levels. In vitro, SW significantly promoted the proliferation of osteoblast cells and the release of ALP, reduced the TRAP-positive cells in osteoclast cells, and modulated the levels of TNF-α, IL-6, IL-1β, MMPs, and NF-κB p65 release in the osteoblast and osteoclast co-culture system. SW also downregulated the expression of TRAP, RANKL, and receptor activator of NF-κB (RANK) levels and upregulated the expression of osteoprotegerin (OPG) levels in vivo and vitro. These results suggested that SW prevents bone erosion by regulating RANKL/RANK/OPG signaling to control the release of pro-inflammatory cytokines and pro-angiogenic markers [101]. In adjuvant-induced arthritic animals, SW treatment modulated red blood cell (RBC), white blood cell (WBC), and erythrocyte sedimentation rate (ESR) levels, reduced the release of acute phase proteins, lysosomal enzymes, protein-bound carbohydrates, and urinary degradation products, and elevated the levels of anti-inflammatory factors IL-4 and IL-10. SW treatment also significantly inhibited the release of NF-κB p65, phospho-inhibitor of kappa B alpha (p-IκBα), phospho-Janus kinase 2 (p-JAK2), and p-STAT3 levels in both arthritic animals and LPS-induced RAW 264.7 macrophage cells. These results indicated that the anti-arthritic effect of SW is closely related to NF-κB/IκB and JAK2/STAT3 signaling [102]. The above studies also revealed that SW might have the potential to treat RA. We summarize the multiple mechanisms by which SW exerts bone- and joint-protective effects in Figure 9.

### 3.8. Prostate-Protective Effects

SW also has significant prostate-protective effects. Benign prostatic hyperplasia (BPH) is a common prostate disease in middle-aged and older men [103]. In testosterone-induced BPH rats, treatment with SW (16 mg/kg/d, p.o.) for 28 days significantly reduced propidium iodide (PI) levels and prostatic acid phosphatase (PACP) activity, as well as ameliorated testosterone-induced morphological alterations, such as collagen deposition and epithelial cell expansion in the prostate. In addition, treatment with SW significantly alleviated testosterone-induced overexpression of vascular endothelial growth factor (VEGF), epidermal growth factor (EGF), basic fibroblast growth factor (βFGF), proliferating cell nuclear antigen (PCNA), androgen receptor (AR) estrogen receptor alpha (ER-α), and hypoxia-inducible factor 1 alpha (HIF-1α) and reduced the ratio of Bcl-2/Bax. SW also obviously increased the levels of total sulfhydryl (T-SH), GSH, COX-2, and SOD and decreased the activity of oxidized glutathione (GSSG) and MDA. SW treatment downregulated Twist, Vimentin, and α-SMA levels and upregulated E-cadherin levels, thereby attenuating prostatic EMT. These results suggest that SW has anti-prostatic hyperplasia, anti-inflammatory, and antioxidant effects [104]. In vivo, SW (32 and 16 mg/kg/d) relieved chronic cigarette smoke (CS)-induced prostatic collagen deposition and morphological changes by inhibiting the expression levels of Hyp, collagen type I (Col-I), and Col-III in rats. SW also significantly ameliorated CS-induced prostate local inflammation by decreasing the levels of pro-inflammatory mediators (IL-1β, IL-6, TNF-α, and NO) and the activities of COX-2 and iNOS; it also relieved oxidative stress by elevating the activities of antioxidant enzymes (SOD, CAT, and GPx), increasing the levels of TAOC, T-SH, and GSH, and decreasing the contents of MDA and GSSG. In vitro, SW inhibited CS-induced proliferation of human prostate RWPE-1 and WPMY-1 cells. SW also significantly reduced the expression levels of α-SMA, Indian hedgehog (IHH), smoothened (SMO), glioma-associated oncogene homolog (GLI)-1, Snail, zinc finger E-Box binding Homeobox 1 (ZEB1), and TGF-β1 while elevating the levels of E-cadherin in rat prostate tissues and human prostate cells, thereby inhibiting the prostatic epithelial–mesenchymal transition (EMT) and the activation of the HH signaling pathway to ameliorate CS-induced prostatic fibrosis [105]. In Figure 10, we summarize the multiple mechanisms by which SW exerts a prostate-protective effect.

### 3.9. Other Therapeutic Effects

Radiation-induced intestinal injury (RIII) is one of the intestinal complications caused by radiotherapy for abdominal or pelvic tumors [106]. A study showed that SW has significant protective effects against RIII both in vitro and in vivo. In radiation-induced cellular and mouse intestinal injury models, prophylactic administration of SW attenuated radiation damage to cells by reducing ROS and superoxide anion levels. SW also increased the survival rate, improved the morphological injury of the small intestine, and reduced the relative abundance of primary bile acids in irradiated mice. Further research elucidated that SW could prevent RIII by reducing DNA damage and inhibiting the activation of the cGAS-STING (the cyclic GMP-AMP synthase-Stimulator of interferon genes) pathway [107]. SW exhibited both peripheral and central antinociceptive activity dose-dependently in three different mouse pain models [24]. Molecular docking studies also showed that SW has good binding interactions with the cyclooxygenase-2 protein [108].

SW showed significant antibacterial [109] and antifungal activity against *P. aeruginosa*, *S. aureus*, and *Trichoderma viride*. The minimum inhibitory concentrations (MICs) of SW for *S. typhi* and *K. pneumoniae* were 1 pg/mL [28,110]. In zebrafish infected with *Salmonella typhi*, SW significantly reduced the bacterial load in zebrafish, attenuated the symptoms of infection, such as spinal deformity, and reduced mortality; therefore, it may be a promising anti-typhoid drug [111].

In human granulosa luteal cells from insulin-resistant polycystic ovarian syndrome patients (PCOS-IR), the administration of SW significantly reversed the expression of insulin resistance-related factors, including insulin receptor (INSR-β), PI(3)K, p-Akt, protein kinase C (PKCζ), PPARγ, p-IRS (Ser307), and insulin-like growth factor 1 (IGF) systems, as well as modulated lipid metabolism-related genes, re-established steroidogenesis, and restored 17-beta hydroxy steroid dehydrogenase (17β-HSD) and 3-beta hydroxy steroid dehydrogenase (3β-HSD) enzyme activities and hormone levels, indicating that it can be used as an insulin sensitizer to alleviate PCOS-IR [112]. SW stimulated gastric emptying and gastrointestinal motility by inhibiting dopamine D2 receptor, suggesting that it may be a potential treatment for functional dyspepsia [27]. In TGF-β1-stimulated A549 cells, swertiamarin downregulated the expression levels of α-SMA, E-cadherin, lysyl oxidase (LOX), collagen type V alpha 2 chain (COL5A2), and connective tissue growth factor (CTGF). This suggests that SW exerts an anti-pulmonary fibrosis effect by inhibiting the EMT process [113,114].

SW has significant cardioprotective effects. In isoproterenol-induced myocardial infarction (MI) rats, pre-treatment with SW notably alleviated myocardial fibrosis and inflammatory cell infiltration. It also restored the levels of several cardiac marker enzymes, oxidative stress markers, antioxidant enzymes, Na^+^/K^+^ and Ca^2+^ ATPases, and pro-inflammatory cytokines [115].

## 4. Toxicology

The toxicity of SW has not been adequately and systematically studied. The available studies suggest that it is relatively safe at appropriate doses, but more data are needed to fully assess its toxicity for long-term use. Based on toxicity studies in Wistar rats, SW demonstrated excellent safety with no mortality or adverse effects observed at doses up to 2000 mg/kg in acute studies (LD_50_ > 2000 mg/kg) and up to 500 mg/kg daily for 50 days in subchronic studies. No significant changes were found in hematological parameters, biochemical markers, or histopathological examination of vital organs (liver, kidney, pancreas), indicating SW has a wide safety margin and low toxicity potential suitable for chronic therapy [116].

A toxicity study of SW in zebrafish showed low or no toxicity at low concentrations (40 µM), while high doses (243 µM) caused organ malformations and developmental abnormalities in zebrafish embryos, as well as significant changes in zebrafish liver enzymology and antioxidant activity [117]. Swertiamarin exhibited significant general toxicity in the brine shrimp lethality bioassay, and the LD50 value was 8.0 microg/mL [109].

## 5. Traditional Medical Background and Application of Swertiamarin

Swertia plants (genus *Swertia*, *Gentianaceae* family) have been historically utilized in multiple traditional medical systems, including Ayurveda, Traditional Chinese Medicine (TCM), and Tibetan medicine. Swertiamarin, an iridoid glycoside derived from the genus *Swertia*, serves as the core bioactive component underpinning the applications in these traditional medical systems.

### 5.1. Applications in Ayurvedic Medicine

*Swertia chirayita*, known as “Kirata-tikta” in Ayurvedic medicine, has been used for centuries. The dried whole herb has been widely used for the treatment of fever, indigestion, and “Madhumeha” (a symptom of diabetes described in ancient Indian texts). Swertiamarin, its main active ingredient, manifests intensely bitter properties. In Ayurvedic theory, this bitter substance is believed to possess the properties of cooling, detoxification, and regulation of liver and gallbladder functions, and it is often used to balance the inflammatory state caused by excess “Pitta” (fire) energy [118,119]. Ayurvedic preparations containing *Swertia chirayita* have been utilized in the treatment of various diseases. Ayush-64, which possesses antiviral and immunomodulatory properties, has been employed as an adjunct to standard care for patients with mild to moderate COVID-19 [120].

### 5.2. Applications in Traditional Chinese Medicine (TCM) and Other Traditional Medicine

In the Traditional Chinese Medicine (TCM) system, plants of the genus *Swertia* are known as “Dangyao (*Swertia pseudochinensis* Hara)” or “Zangyinchen (*Swertia mussotii* Franch)” [121]. Long-Dan-Xie-Gan-Tang (LDXGT) is a well-known traditional Chinese medicine formula. *Gentiana scabra* (containing swertiamarin) is one of the main ingredients in this formula. LDXGT is used in TCM for the treatment of chronic hepatitis, jaundice, cystitis, and eczema [31]. The combination of LDXGT with psychotropic medications enhanced efficacy and reduced certain adverse effects associated with psychotropic medications [122]. An oral *Gentiana* formula also provided pain relief and reduced the incidence of postherpetic neuralgia [123]. *Swertia mussotii* Franch has been used as a traditional Tibetan medicine for thousands of years, and it is rich in the active ingredient SW.

## 6. Conclusions and Future Perspectives

The pharmacokinetic profile of SW in rats is characterized by rapid absorption, low bioavailability, and rapid metabolism and clearance. It is mainly metabolized in the intestinal tract and the liver, with its metabolites excreted through the urine. SW is affected by some factors, such as gender, dosing combination, and dosage, and thus exhibits different pharmacokinetic profiles. And, we found an interesting difference: the same SD rat strain, similar assay method (LC-MS/MS), but different bioavailability (10.3% at 25 mg/kg, 8.0% at 50 mg/kg, 6.7% at 100 mg/kg, and 6.2% at 150 mg/kg) [30,41]. The possible reasons for this difference include (1) bioavailability is affected by the dose factor, and there is a saturation effect; (2) different pre-treatment methods (protein precipitation versus solid-phase extraction); and (3) differences in mahunpectrometry detection modes (positive versus negative ion modes). To overcome the challenge of the low bioavailability of SW, future researchers could focus on investigating the therapeutic efficacy of co-administration and alternative routes of SW administration, such as sublingual and transdermal administration, to explore the possibilities of the clinical translation of SW in the future.

In addition, semi-synthetic analogues of SW have also become a new direction of investigation with the continuous advancement of research. Modifications to chemical structures can further enhance their biological activities and metabolic properties in specific aspects. Among several semi-synthetic SW analogues, these compounds showed higher docking scores for different GLUTs and better anti-diabetic effects in terms of glucose uptake and insulin secretion in the NIT-1 cell line compared to Glibenclamide and SW [124]. Additionally, several other SW analogues have been shown to inhibit dipeptidyl peptidase IV (DPP-IV) enzyme activity, indicating their potential as targeted anti-diabetic agents [125]. Moreover, several new glycosides also exhibit anti-HBV activity [126].

As an active compound of natural origin, SW shows potential hepatoprotection, hypoglycemia, neuroprotection, and anti-tumor therapeutic effects through multi-level, multi-target, and multi-mechanism processes (Figure 11 and Figure 12). Oxidative stress and inflammatory response are closely related to a variety of chronic diseases. SW can regulate classical signaling pathways, such as NF-κB, MAPK, and PI3K/Akt, to exert anti-inflammatory and antioxidant effects, thus achieving multiple biological effects. For example, SW can exert hepatoprotective and neuroprotective effects by inhibiting Nrf2-mediated oxidative stress. SW can ameliorate APAP-induced liver injury, OGDR-stimulated human neuronal SH-SY5Y cell injury, obesity-associated chronic inflammation, and arthritis through the NF-κB signaling pathway; SW also attenuates the LPS-induced fever and insulin resistance by regulating PI3K-Akt and p38 MAPK pathways.

Notably, although SW has significant anti-inflammatory and antioxidant activities, its specific effects on skin injury repair remain unexplored. Future studies should evaluate its potential in wound healing models, particularly given its established modulation of TGF-β, NF-κB, and Nrf-2 pathways in other tissues, such as the hepatic and renal systems.

Additionally, SW plays a dual role in promoting or inhibiting apoptosis. For example, SW can promote apoptosis in HepG2 hepatocellular carcinoma cells, human peripheral blood mononuclear cells (PBMCs), and neuroblastoma (SK-N-AS) cell lines to exert anti-tumor effects. On the other hand, it inhibits apoptosis in hepatocytes to ameliorate hepatic injury and inhibits I/R-induced neuronal apoptosis to exert neuroprotective effects.

This review summarizes the pharmacokinetics, pharmacological effects, and molecular mechanisms of SW, but its clinical translation is still subject to multiple limitations: (1) The low oral bioavailability, short half-life, and rapid metabolism properties of SW are important challenges to therapeutic efficacy. (2) The toxicological evaluation of SW is still insufficient, especially for long-term toxicity and reproductive toxicity studies in mammals. (3) Currently, the data are from preclinical studies (animals/cells), and how to translate in vitro concentrations to human conditions is also a question for future research. (4) There is a lack of in-depth studies on SW’s targets as well as on the intricate interactions between signaling pathways. (5) Although swertiamarin-containing traditional preparations are used in medicine for the treatment of a wide range of diseases, the exploration of the single active ingredient is still not comprehensive. As SW is a potential natural lead compound, in the future, we need to address the bottleneck of pharmacokinetics, optimize target selection, tap into the wisdom of ethnomedicine, and use artificial intelligence for precision drug design. Only when these problems are solved can SW be transformed from a “promising natural compound” into a clinically viable drug and hopefully an adjunctive therapeutic agent for various chronic diseases.

## Figures and Tables

**Figure 1 cells-14-01173-f001:**
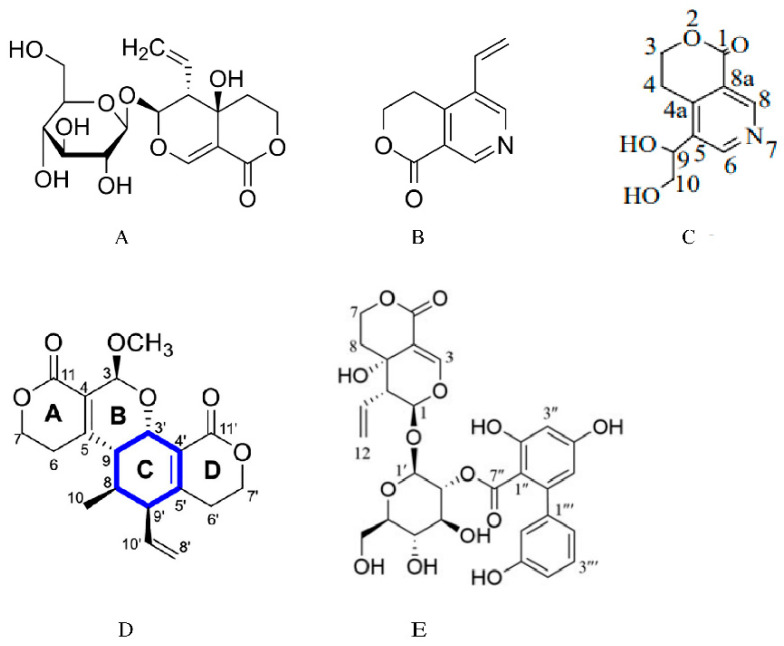
Chemical structure of swertiamarin, and the structures of its metabolites and analogues. (**A**) Chemical structure of swertiamarin. (**B**) Chemical structure of gentianine [14]. Swertiamarin is metabolized to gentianine by intestinal microorganisms. (**C**) Chemical structure of gentiandiol [14]. Gentianine is further converted to the nitrogen-containing metabolite gentiandiol in the liver. (**D**) Chemical structure of sweritranslactone D [15]. Sweritranslactone D is a hepatoprotective novel secoiridoid dimer with a tetracyclic lactone skeleton from heat-transformed swertiamarin. (**E**) Chemical structure of swertiamarin B [16], Swertiamarin B, a structural analogue of SW, is isolated from the ethanol extract of the aerial parts of *Swertia mussotii*.

**Figure 2 cells-14-01173-f002:**
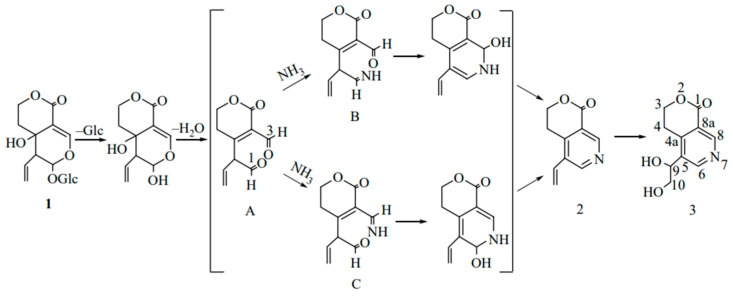
Proposed metabolic pathways for the conversion of swertiamarin to nitrogen-containing compounds in vivo [14].

**Figure 3 cells-14-01173-f003:**
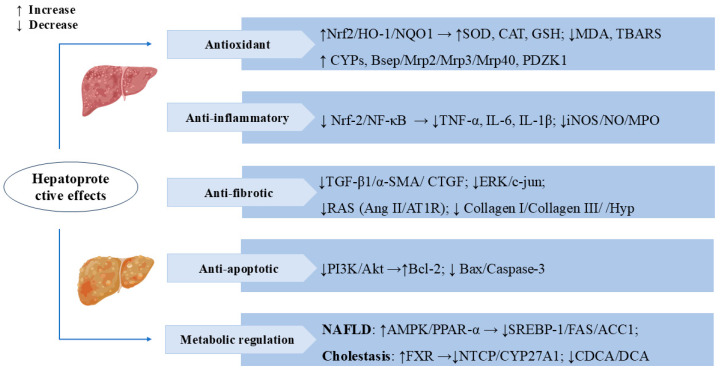
Swertiamarin exerts multifaceted hepatoprotective effects by targeting key molecular pathways. (1) Antioxidant: SW activates the Nrf2/HO-1/NQO1 pathway to enhance antioxidant enzymes (SOD, CAT, GSH) and reduce oxidative stress markers (MDA, TBARS). It upregulates cytochrome P450 (CYPs) and bile transport proteins, promoting toxin metabolism and excretion. (2) Anti-inflammatory: SW suppresses the Nrf-2/NF-κB signaling pathway, thereby decreasing the release of pro-inflammatory cytokines (TNF-α, IL-6, IL-1β) and reducing the expression of inflammatory mediators (iNOS/NO/MPO). (3) Anti-fibrotic: SW inhibits the TGF-β1/α-SMA/CTGF signaling cascade, leading to reduced collagen deposition (collagen I/III) and Hyp levels, and blocks the RAS system (Ang II/AT1R) and ERK/c-jun signaling, thereby slowing the progression of hepatic fibrosis. (4) Anti-apoptotic: SW modulates the PI3K/Akt pathway to upregulate the anti-apoptotic protein Bcl-2 while downregulating pro-apoptotic factors (Bax/caspase-3). (5) Metabolic regulation: SW improves NAFLD via AMPK/PPAR-α/SREBP-1 and alleviates cholestasis by activating FXR, inhibiting NTCP/CYP27A1 and toxic bile acids (CDCA/DCA).

**Figure 4 cells-14-01173-f004:**
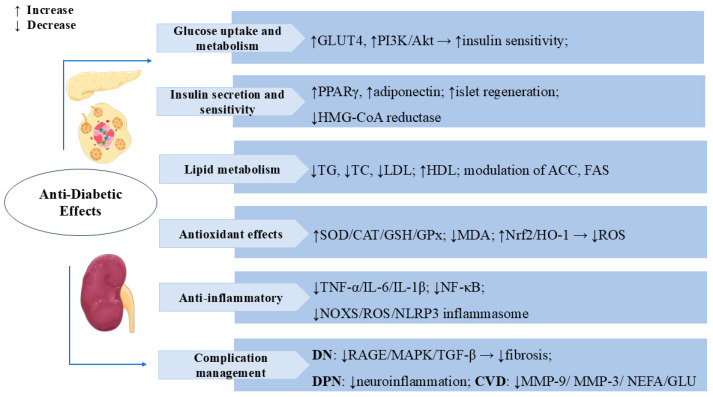
Swertiamarin exerts comprehensive anti-diabetic effects through multiple mechanisms: (1) Glucose uptake and metabolism: SW enhances glucose metabolism by promoting GLUT4 expression and activating the PI3K/Akt pathway to improve insulin sensitivity while stimulating pancreatic islet regeneration. (2) Insulin secretion sensitivity: SW improves insulin secretion and sensitivity by upregulating PPARγ and adiponectin expression and inhibiting HMG-CoA reductase activity. (3) Lipid Metabolism: SW modulates lipid metabolism by reducing TG, TC, and LDL levels while increasing high-density lipoprotein (HDL) and regulates key enzymes, including ACC and FAS. (4) Antioxidant Effects: SW elevates SOD, CAT, GSH, and GPx while decreasing MDA through Nrf2/HO-1 pathway activation to reduce ROS. (5) Swertiamarin demonstrates anti-inflammatory effects by suppressing TNF-α, IL-6, IL-1β, NF-κB signaling, and the NOX4/ROS/NLRP3 inflammasome. (6) Complication management: SW manages diabetic complications by inhibiting RAGE/MAPK/TGF-β pathways, reducing neuro-inflammation, MMP-9, MMP-3, NEFA, and glucose levels.

**Figure 5 cells-14-01173-f005:**
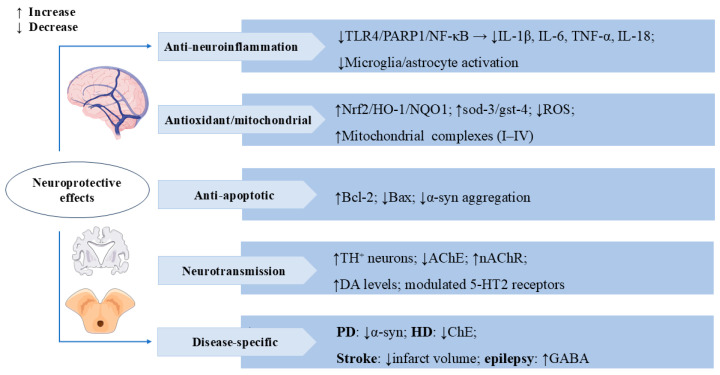
Swertiamarin exerts multi-target neuroprotective effects through anti-neuro-inflammatory, antioxidant, and neuromodulatory mechanisms. (1) Anti-neuro-inflammatory: SW inhibits TLR4/PARP1/NF-κB signaling to reduce pro-inflammatory cytokines and suppresses microglia/astrocyte activation. (2) Antioxidant/Mitochondrial: SW demonstrates potent antioxidant activity by activating the Nrf2/HO-1/NQO1 pathway, upregulating sod-3/gst-4, reducing ROS, and enhancing mitochondrial complexes I-IV. (3) Anti-apoptotic: SW upregulates Bcl-2 and downregulates Bax while inhibiting α-synuclein aggregation. (4) Neurotransmission: SW modulates neurotransmission by increasing tyrosine hydroxylase-positive (TH+) neurons and DA levels, inhibiting AChE and nAChRs, and regulating 5-HT2 receptors. (5) Disease-specific benefits include reducing α-synuclein in Parkinson’s disease (PD), inhibiting cholinesterase (ChE) in Huntington’s disease (HD), decreasing infarct volume in stroke, and enhancing GABA activity in epilepsy.

**Figure 6 cells-14-01173-f006:**
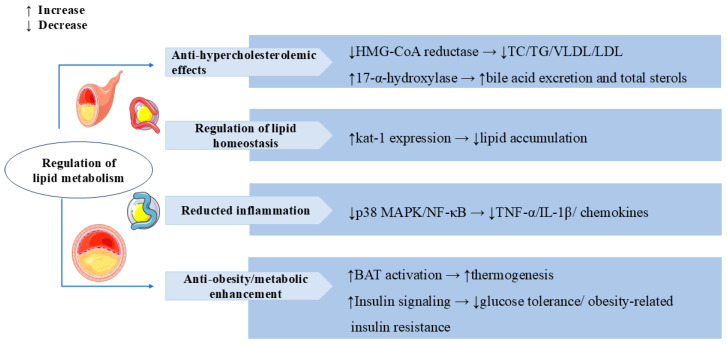
Swertiamarin demonstrates comprehensive lipid-modulating and metabolic regulatory effects through multiple mechanisms. (1) Anti-hypercholesterolemic: SW inhibits HMG-CoA reductase to reduce total cholesterol (TC), triglyceride (TG), and LDL/VLDL levels while simultaneously enhancing bile acid excretion through 17-α-hydroxylase activation. (2) Regulation of lipid homeostasis: SW reduces lipid accumulation by elevating the kat-1 level. (3) SW reduces pro-inflammatory mediators and chemokines by suppressing p38 MAPK/NF-κB pathways. (4) Anti-obesity and metabolic enhancement: SW increases the activation of BAT to promote thermogenesis and ameliorates glucose intolerance and obesity-related insulin resistance by enhancing insulin signaling.

**Figure 7 cells-14-01173-f007:**
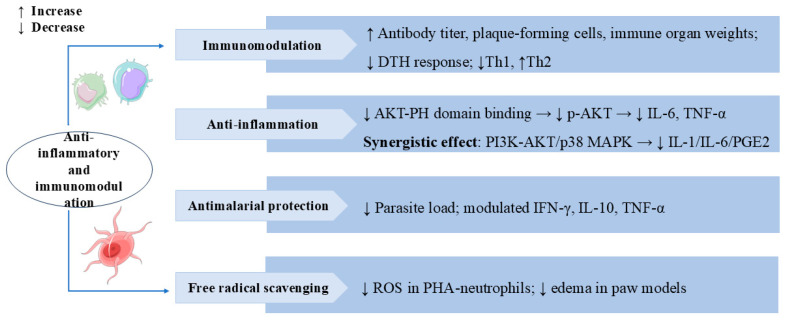
Swertiamarin exhibits broad immunomodulatory and anti-inflammatory properties through multiple mechanisms. (1) Immunomodulation: SW enhances humoral and cellular immunity by increasing antibody titers, plaque-forming cells, and immune organ weights and reducing delayed-type hypersensitivity (DTH) responses and Th1/Th2 balance shifting. (2) Anti-inflammation: SW inhibits AKT-PH domain binding to suppress p-AKT activation, consequently reducing pro-inflammatory cytokines, and synergistically modulates both the PI3K-AKT and p38 MAPK pathways to decrease IL-1, IL-6, and PGE2 production. (3) Antimalarial protection: SW reduces parasite load and regulates cytokine profiles for antimalarial effects. (4) Free radical scavenging: SW reduces ROS in PHA-stimulated neutrophils and edema in paw models.

**Figure 8 cells-14-01173-f008:**
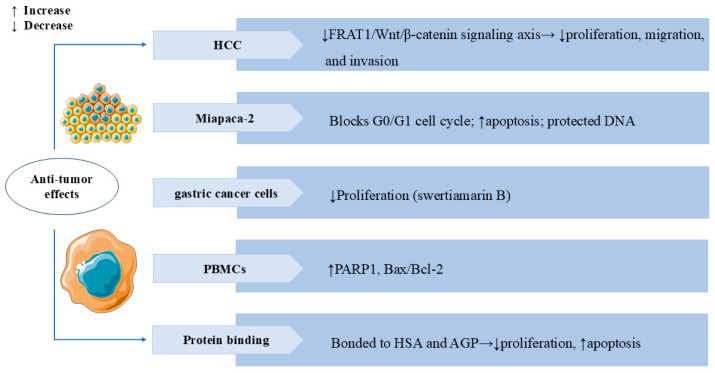
Swertiamarin and its analogues exhibit broad anti-tumor activity across multiple cancer types. (1) HCC: SW inhibits tumor proliferation, migration, and invasion by suppressing the FRAT1/Wnt/β-catenin signaling axis. (2) Miapaca-2: SW induces G0/G1 cell cycle arrest and promotes apoptosis while protecting normal DNA integrity. (3) Gastric cancer cells: Swertiamarin B demonstrates significant antiproliferative effects. (4) PBMCs: SW upregulates pro-apoptotic markers, including PARP1 and the Bax/Bcl-2 ratio. (5) Protein binding: SW binds to HSA and AGP to inhibit proliferation and induce apoptosis in SK-N-AS cells.

**Figure 9 cells-14-01173-f009:**
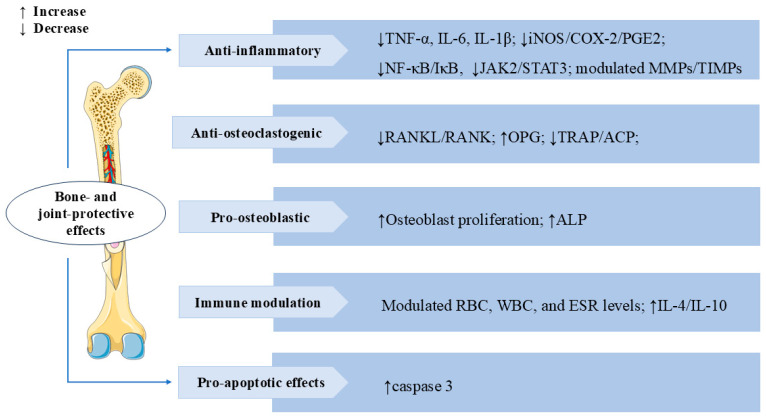
Swertiamarin demonstrates comprehensive bone-protective and immunomodulatory effects through multi-target mechanisms. (1) Anti-inflammatory: SW suppresses pro-inflammatory cytokines (TNF-α, IL-6, IL-1β) and iNOS/COX-2/PGE2 production and downregulates the NF-κB/IκB and JAK2/STAT3 signaling pathways while modulating MMPs/TIMPs balance. (2) Anti-osteoclastogenic: SW inhibits osteoclast formation by reducing RANKL/RANK signaling and TRAP/ACP activity. (3) Pro-osteoblastic: SW promotes osteoblast growth and ALP expression. (4) Immune modulation and pro-apoptotic effect: SW modulates immune function by normalizing RBC/WBC counts and ESR levels while enhancing anti-inflammatory cytokines (IL-4/IL-10) and activating caspase-3-mediated apoptosis.

**Figure 10 cells-14-01173-f010:**
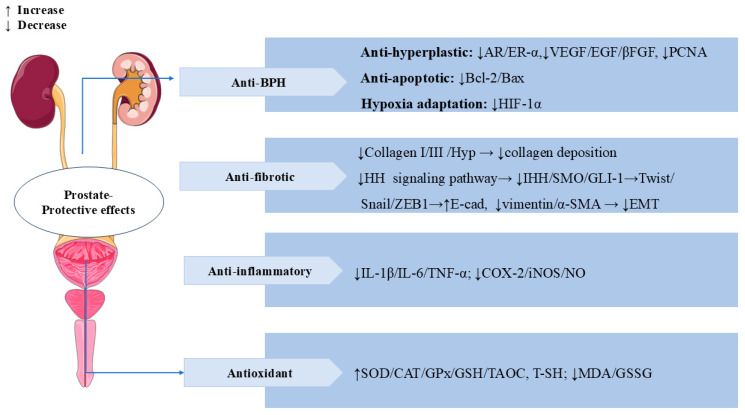
Swertiamarin demonstrates multi-target therapeutic effects for prostate disorders through four key mechanisms. (1) BPH: SW exerts anti-hyperplastic action by modulating androgen/estrogen receptors (AR/ER-α), growth factors (VEGF/EGF/βFGF), and cell proliferation markers (PCNA), balances apoptosis (Bcl-2/Bax ratio), and inhibits hypoxia adaptation (HIF-1α). (2) Anti-fibrotic protection: SW reduces collagen deposition and prevents EMT by the HH pathway. (3) Anti-inflammatory and antioxidant: SW lowers pro-inflammatory mediators while enhancing antioxidant defenses and reducing oxidative stress.

**Figure 11 cells-14-01173-f011:**
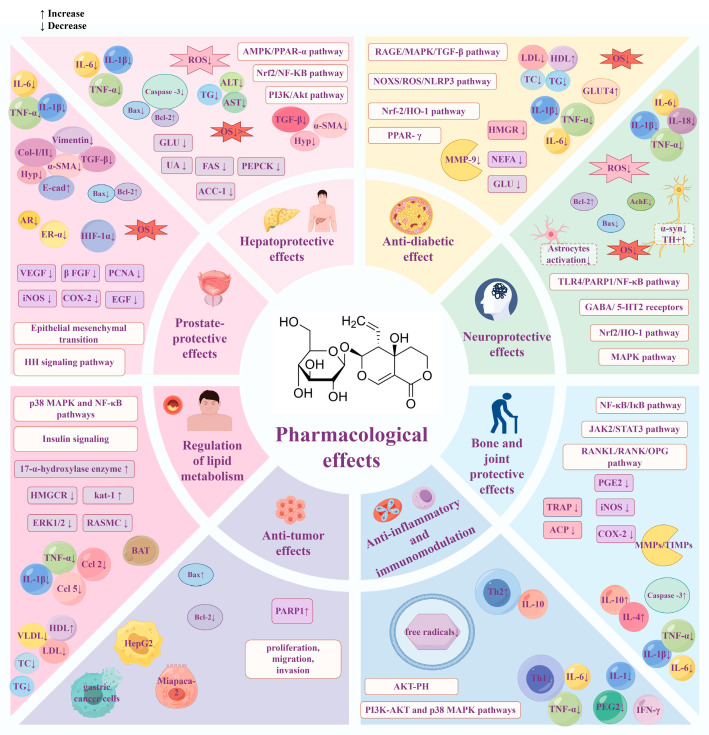
Summary of the pharmacological effects of swertiamarin. SW exhibits a broad spectrum of pharmacological activities, including hepatoprotection, lipid metabolism regulation, neuroprotection, anti-inflammatory effects, anti-tumor effects, anti-diabetic effects, prostate protection, and bone and joint protection. Created with https://www.figdraw.com (accessed on 22 May 2025).

**Figure 12 cells-14-01173-f012:**
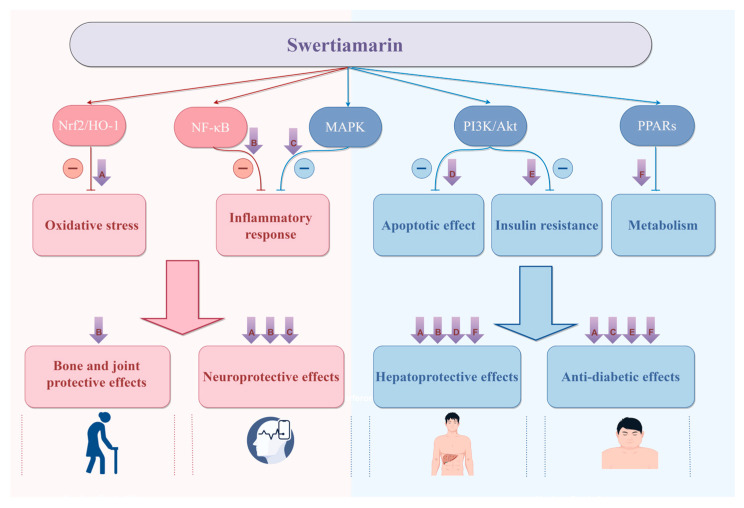
There is a potential connection between different biological effects of SW, which can exert multiple pharmacological effects through multiple targets and multiple mechanisms, mainly including (1) SW can target Nrf2/HO-1 to inhibit oxidative stress to exert hepatoprotective, anti-diabetic, and neuroprotective effects. (2) SW can target NF-κB to inhibit inflammation to exert hepatoprotective, bone- and joint-protective, and neuroprotective effects. (3) SW can target MAPK to inhibit inflammation to exert neuroprotective and anti-diabetic effects. (4) SW can target PI3K and Akt to inhibit apoptosis to exert hepatoprotective effects and inhibit insulin resistance to exert anti-diabetic effects. (5) SW can target PPARs to regulate metabolism to exert hepatoprotective and anti-diabetic effects. Created with https://www.figdraw.com (accessed on 22 July 2025). All abbreviations are listed in Appendix A.

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
