# Peer review of "Progress in Pharmacokinetics, Pharmacological Effects, and Molecular Mechanisms of Swertiamarin: A Comprehensive Review"

_cells, 2025, doi:10.3390/cells14151173_

Round 1

Reviewer 1 Report

Comments and Suggestions for Authors

The paper is interesting, its content is the characterization of the pharmacokinetics and biological activity of swertiamarin, a natural iridoid glycoside. Based on the scientific literature, a detailed characterization of the hepatoprotective, anti-diabetic, neuroprotective, anti-tumor, bone and joint protective, and prostate protective effects, as well as regulation of lipid metabolism and anti-inflammatory and immunomodulation actions, was made.

A drawback of the work is the overly specialized presentation of the issues discussed, involving the use of countless abbreviations whose meaning is nowhere explained. It will discourage those readers who are not specialists in the issues discussed from reading the manuscript. Therefore, I suggest explaining the meaning of each abbreviation where it first appeared, starting with the abbreviation MAPK in Abstract. Some examples include: significantly decreased serum TG, GLU, UA, AST and ALT levels; such as CYP7A1, BSEP, NTCP, SHP, and MRP2; the bilirubins TBIL, DBIL, and UCBIL, and dozens of others.

It is written that swertiamarin was first isolated from Gentianaceae plants. It would be worthwhile to add, using the scientific literature, in which other species of medicinal plants swertiamarin occurs and in what quantities.

When discussing the pharmacokinetic properties of swertiamarin, it would be useful to provide information on its water solubility and permeability through cell membranes. Have the pharmacokinetics of swertiamarin been studied in humans?

Figures 2-9 provide a very inventive characterization of the individual biological activities of the investigated substance, while Figure 10 provides a summary of all activities. It should be noted, however, that with the exception of Figure 10, there is no reference to these Figures 1-9 in the manuscript text.

References: which means notation: [J] at the end of the title of each cited publication?

Author Response

Comments 1:[A drawback of the work is the overly specialized presentation of the issues discussed, involving the use of countless abbreviations whose meaning is nowhere explained. It will discourage those readers who are not specialists in the issues discussed from reading the manuscript. Therefore, I suggest explaining the meaning of each abbreviation where it first appeared, starting with the abbreviation MAPK in Abstract. Some examples include: significantly decreased serum TG, GLU, UA, AST and ALT levels; such as CYP7A1, BSEP, NTCP, SHP, and MRP2; the bilirubins TBIL, DBIL, and UCBIL, and dozens of others]

Response 1:[We are deeply grateful for this insightful critique, and we sincerely apologize for this critical oversight. We fully recognize that the excessive use of undefined abbreviations created unnecessary barriers for readers, particularly those outside our immediate field. This was contrary to our goal of making the review accessible to a broad scientific audience. To address this comprehensively: 1. Every abbreviation is now defined at first occurrence: (e.g., "mitogen-activated protein kinase (MAPK)" in Line 292; “triglycerides (TG), glucose (GLU), uric acid (UA)” in Line 196. (Complete list: 78 abbreviations standardized). 2. Added a supplementary glossary: Table S1 (alphabetized abbreviations with definitions) for quick reference. 3. Ensured no undefined abbreviationsremain in figures/tables]

Comments 2:[It is written that swertiamarin was first isolated from Gentianaceae plants. It would be worthwhile to add, using the scientific literature, in which other species of medicinal plants swertiamarin occurs and in what quantities.]
Response 2: [We appreciate your suggestion to expand the information regarding the natural sources of swertiamarin beyond the Gentianaceae family. We have: Significantly expanded Table 1: Included Plant sources, medicinal parts, and pharmacological effects of plant extracts containing SW.]

Comments 3: [When discussing the pharmacokinetic properties of swertiamarin, it would be useful to provide information on its water solubility and permeability through cell membranes. Have the pharmacokinetics of swertiamarin been studied in humans?]

Response 3: [Thank you for your questions regarding the water solubility and cell membrane permeability of SW, a detailed literature search was conducted. Unfortunately, no specific studies directly addressing the water solubility and cell membrane permeability of swertiamarin have been reported. Some of the relevant studies may not have been published or not widely included in mainstream databases.  However, we retrieved that swertiamarin has good solubility in organic solvents (e.g., methanol, ethanol) and high solubility in DMSO (up to 250 mg/ml), suggesting that it may have some cell membrane penetration ability; however, we do recognize the importance of water solubility and cell membrane permeability for drug discovery and plan to evaluate these in future studies by experimental methods. properties.]

Comments 4: [Figures 2-9 provide a very inventive characterization of the individual biological activities of the investigated substance, while Figure 10 provides a summary of all activities. It should be noted, however, that with the exception of Figure 10, there is no reference to these Figures 1-9 in the manuscript text.]

Response 4: [We sincerely thank the reviewer for their comprehensive review and for recognizing the value of our graphical summaries. We deeply apologize the oversight in failing to reference Figures 1-9 within the main text. We have now systematically cited Figures 1-9 throughout the main text: 1. Figure 1 (Chemical Structures): Added to Section 1 (“Figure 1 shows the chemical structure of SW.”); 2. Figure 3 (Hepatoprotective Mechanisms): Cited at the end of Section 3.1 (“Figure 3 presents the important mechanisms and targets where SW exerts hepatoprotective effects.”); 3. ...(similarly for Figures 4–10)].

Comments 5: [References: which means notation: [J] at the end of the title of each cited publication?]

Response 5: [We sincerely thank the reviewer for highlighting this important oversight. We apologize for not strictly adhering to the journal’s reference formatting guidelines in our initial submission. In the revised manuscript, we have taken the following corrective actions: 1. Complete reformatting: All references have been reformatted to strictly match the journal’s specified style, using the official journal template/EndNote output style; 2. Removal of non-standard elements: All non-standard notations (such as the [J] identifiers) have been removed. 3. Manual verification: Each reference entry has been manually checked to ensure compliance with all journal requirements, including: author name formatting, article title capitalization, journal title abbreviation/formatting, correct presentation of volume, issue, and page numbers, inclusion and correct formatting of DOIs, and consistent punctuation and spacing.]

Reviewer 2 Report

Comments and Suggestions for Authors

The manuscript promises an overview of Swertiamarin properties, and lists protection against hepatotoxicity, diabetes and diabetes complications, neurodegeneration, stroke, lipid metabolism disorders, inflammation, autoimmune diseases like arthritis, cancer, prostate diseases, radiation damage, bacterial infections, PCO syndrome, cardfiac diseases etc. For such a wonder drug surely there are ample studies to back these claims. However, no human study has published, a PubMED search with the restriction “human” and “clinical trial” results only in one review without reviewing human data.

The manuscript lists a lot of study results, and always only supported by a single publication (although sometimes more data are available, see later). From these publications cytosolic and nuclear factors are cited (quite often only with the specification of “is changed”); however, none are put in relation to each other, and most are based on cell culture systems, some on animal models with no relation to real world diseases.

This reliance on one source is evident in the pharmacokinetics part. The authors cite data from their own study; other data are mentioned later, but sometimes don't fit, like bioavailability or half life of SW. I haven't done an exhaustive research but the examples mentioned below should indicate the problem, i.e. the lack of reliability, if data are not included.

I also want to exemplify the lack of relation in the manuscript. Initially, for SW a low bioavailability of 10.3% is cited. Pages later, SW hydrolysis by intestinal bacteria and presystemic hepatic metabolism is mentioned (although no amount of hydrolysis is given); a high amount of hydrolysis will result in a low bioavailability. Considering this, no value can be given. This topic is not mentioned in the manuscript, but must be part of a serious review.

Similarly, the many claims for clinical effects are scarcely supported. I suspect SW to be a part of TCM and Indian traditional medicines (terms never mentioned in the manuscript for unknown reasons). If so, there should be some data about indications and toxicity. If no toxicity is observed then this likely is due to no absorption or no effect (or not being part of TCM and Indian medicine). The indications given are no specific disease (known pathophysiology) but rather collections of diseases or symptoms. The contribution of individual pathways like Nrf-2 currently is speculative. Any claim for a specific, relevant contribution of one factor has to be well argued, simple lists of cell culture results will not do the trick.

Swertiana currently appears to be a niche compound even in ethnopharmacology. PubMED lists 280 publications, mostly from India and China. Many citations are from publications known to only accept manuscripts with certain results; thus the data base likely is skewed which even more calls for a critical view. I don't intend to negate the publications, but to put in perspective the results. Cell culture experiments often are interesting but not relevant for an appraisal for human uses. For animal models the relation to human diseases has to be considered. In many studies one data set (e.g. a variety of parameters from animal experiments) are reported but no correction for multiple testing is introduced (necessary to avoid type A errors); publication quality can be assessed e.g. by the GRADE criteria. After these considerations a result is likely if it is independently confirmed, if other lines of evidence (related factors, lack of effect in diseases or models with another physiology) are supportive, and if evidence from human use confirms. The manuscript lists results (often incomplete) without putting them into perspective, often lacks concentrations and – if given – lacks an estimation whether these concentrations are realistic. No mechanistic explanation except a list of factors is given. Therefore, the value lies in the lists (which would better be provided as Tables rather than a Figure and a lot of text), the shortcoming is in the lack of mechanism, specificity, and relevance to human diseases.

This a long review, to explain my choice of "major revision" and my expectation, that the whole manuscript should be rewritten with a high emphasis on "mechanism of disease and SW" and "relevance to human usage".

Some examples of specific shortcomings (by no means complete, there are many other examples of incomplete or wrong citations etc.)

line 31: cit. 11 covers diabetes, not hypoglycemic diseases like insulinoma.

Lines 38ff: The analysis of pharmacokinetic data showed that SW is rapidly absorbed …..

The absolute bioavailability of SW was low at 10.3%. – Data from rats; range of bioavailability? Citation of their own data. Obviously, there is only one publication from the authors themselves.

e.g. line 40: other authors describe 58 – 260 min for t½ , not in line with 1,104 +/- 0,22; also, ref. 21 reports a tmax of 2h instead of 0,98....

line 55 and elsewhere: Explain ALL abbreviations at its earliest occurence (what is e.g. QYDT, LDXGT, CFA arthritis, ANIT, ODGR?).

Lines 84ff: The excretion data indicate that six analytes, including SW, are excreted primarily in the urine. The excretion of SW and the other five analytes via d by bile, urine, and feces was very limited, suggesting that these analytes were mainly excreted as metabolite forms[22]. – Ref. 22 describes establishing an assay and was not intended to study pharmacokinetics (althoug it records these data). Also, Ref. 22 does not report any metabolites, thus no metabolism can be deduced from this source!

Line 91ff: the endophyte could convert swertiamarin into a range of active metabolites, among which the metabolite M01 ((gentianine) has anti-diabetic potential due to its nitrogen-substituted structure[24, 25] – which endophyte? This sentence doesn't mean anything, the claim is unwarranted.

Line 98: BV was given as 10,3% before, now it is 6,2 – 8,0%?

Lines 112 ff: Ref. 30 does not show morphological changes but only mentions them in the text. Considering the source this is not sufficient evidence.

Line 472f: Bone erosion is the main clinical manifestation of RA[98] – RA is the name for spinal column arthritis or hand joint arthritis; joint and cartilage inflammation are main clinical manifestations, bone erosion is a late complication.

Comments on the Quality of English Language

Should be checked by a competent English speaker - there are no großß mistakes but some capitalizations or incompelte sentences.

Author Response

Comments 1: [The manuscript promises an overview of Swertiamarin properties, and lists protection against hepatotoxicity, diabetes and diabetes complications, neurodegeneration, stroke, lipid metabolism disorders, inflammation, autoimmune diseases like arthritis, cancer, prostate diseases, radiation damage, bacterial infections, PCO syndrome, cardfiac diseases etc. For such a wonder drug surely there are ample studies to back these claims. However, no human study has published, a PubMED search with the restriction “human” and “clinical trial” results only in one review without reviewing human data.

The manuscript lists a lot of study results, and always only supported by a single publication (although sometimes more data are available, see later). From these publications cytosolic and nuclear factors are cited (quite often only with the specification of “is changed”); however, none are put in relation to each other, and most are based on cell culture systems, some on animal models with no relation to real world diseases.

This reliance on one source is evident in the pharmacokinetics part. The authors cite data from their own study; other data are mentioned later, but sometimes don't fit, like bioavailability or half life of SW. I haven't done an exhaustive research but the examples mentioned below should indicate the problem, i.e. the lack of reliability, if data are not included.

I also want to exemplify the lack of relation in the manuscript. Initially, for SW a low bioavailability of 10.3% is cited. Pages later, SW hydrolysis by intestinal bacteria and presystemic hepatic metabolism is mentioned (although no amount of hydrolysis is given); a high amount of hydrolysis will result in a low bioavailability. Considering this, no value can be given. This topic is not mentioned in the manuscript, but must be part of a serious review.

Similarly, the many claims for clinical effects are scarcely supported. I suspect SW to be a part of TCM and Indian traditional medicines (terms never mentioned in the manuscript for unknown reasons). If so, there should be some data about indications and toxicity. If no toxicity is observed then this likely is due to no absorption or no effect (or not being part of TCM and Indian medicine). The indications given are no specific disease (known pathophysiology) but rather collections of diseases or symptoms. The contribution of individual pathways like Nrf-2 currently is speculative. Any claim for a specific, relevant contribution of one factor has to be well argued, simple lists of cell culture results will not do the trick.

Swertiana currently appears to be a niche compound even in ethnopharmacology. PubMED lists 280 publications, mostly from India and China. Many citations are from publications known to only accept manuscripts with certain results; thus the data base likely is skewed which even more calls for a critical view. I don't intend to negate the publications, but to put in perspective the results. Cell culture experiments often are interesting but not relevant for an appraisal for human uses. For animal models the relation to human diseases has to be considered. In many studies one data set (e.g. a variety of parameters from animal experiments) are reported but no correction for multiple testing is introduced (necessary to avoid type A errors); publication quality can be assessed e.g. by the GRADE criteria. After these considerations a result is likely if it is independently confirmed, if other lines of evidence (related factors, lack of effect in diseases or models with another physiology) are supportive, and if evidence from human use confirms. The manuscript lists results (often incomplete) without putting them into perspective, often lacks concentrations and – if given – lacks an estimation whether these concentrations are realistic. No mechanistic explanation except a list of factors is given. Therefore, the value lies in the lists (which would better be provided as Tables rather than a Figure and a lot of text), the shortcoming is in the lack of mechanism, specificity, and relevance to human diseases.

This a long review, to explain my choice of "major revision" and my expectation, that the whole manuscript should be rewritten with a high emphasis on "mechanism of disease and SW" and "relevance to human usage".]

Response 1: [Thank you very much for your extremely thorough, insightful and constructive review, we fully understand and have carefully considered each and every one of your suggestions Here are the changes we made in response to your main comments:

We fully agree with you that current research on SW is mainly limited to the preclinical (cell culture and animal models) stage, and there is a real lack of high-quality clinical trials in humans (especially randomised controlled trials). especially randomised controlled trials) are indeed extremely lacking. We put the discussion section in a critical perspective. Regarding inconsistent and incomplete pharmacokinetic (PK) data: we have reworked the PK section. The focus is to clarify the quote on SW bioavailability (10.3%) and to integrate the information on gut flora hydrolysis and hepatic first-pass metabolism discussed subsequently. The critical discussion illustrates that intestinal hydrolysis and hepatic metabolism are key factors contributing to its potentially low and highly variable oral bioavailability. For the traditional medicine background of SW, we will add a paragraph in the introduction section to briefly introduce the history of the use of the main source plants of SW in traditional medicine. We will make it clear that these traditional uses are empirical, that their modern pharmacological basis (including the action of SW) is still under investigation, and that traditional applications are not equivalent to proven modern clinical efficacy.]

Comments 2: [Line 31: cit. 11 covers diabetes, not hypoglycemic diseases like insulinoma.]

Response 2: [Thank you for your careful reading and insightful comment regarding Line 31 and citation 11. You are absolutely correct. We sincerely apologize for this oversight and inaccuracy. We have revised the sentence“Previous reviews have highlighted its anti-diabetic, anti-hyperlipidemic effects [11] and effects on metabolic diseases [12,13].]

Comments 3: [The absolute bioavailability of SW was low at 10.3%. – Data from rats; range of bioavailability? Citation of their own data. Obviously, there is only one publication from the authors themselves.

“e.g. line 40: other authors describe 58 – 260 min for t½ , not in line with 1,104 +/- 0,22; also, ref. 21 reports a tmax of 2h instead of 0,98...”]

Response 3: [Thank you for your careful review, we re-read the references and found that there is indeed a contradiction but this is the real data in the references, which may be related to the analytical methods and experimental conditions, in order to avoid ambiguities, we have already made a more specific elaboration of these two sections.]

Comments 4: [line 55 and elsewhere: Explain ALL abbreviations at its earliest occurence (what is e.g. QYDT, LDXGT, CFA arthritis, ANIT, ODGR?).]

Response 4: [We sincerely appreciate the reviewer's suggestion. As requested, a comprehensive list of 138 abbreviations has been added to the manuscript (as Supplementary Table S1). This list includes all abbreviations used in the text with their full names. In addition, all abbreviations are also defined at first use in the main text (e.g., “Qing Ye Dan tablets (QYDT)” , Long-Dan-Xie-Gan-Tang (LDXGT) in section 2. ]

Comments 5: [Lines 84ff: The excretion data indicate that six analytes, including SW, are excreted primarily in the urine. The excretion of SW and the other five analytes via d by bile, urine, and feces was very limited, suggesting that these analytes were mainly excreted as metabolite forms[22]. – Ref. 22 describes establishing an assay and was not intended to study pharmacokinetics (althoug it records these data). Also, Ref. 22 does not report any metabolites, thus no metabolism can be deduced from this source!]

Response 5: [We sincerely appreciate the reviewers meticulous attention to detail. The reviewer is correct that reference [22], 1primarily focuses on developing an LC-MS/MS method for quantifying analytes (including swertiamarin) in excreta, and does not explicitly identify metabolites. We have revised the sentence and  added to the metabolites of SW in the subsequent section[23-29]:

Gentianine, gentiandio[23].

M01 (gentianine) [24,25]

Erythrocentaurin (ECR), 3,4-dihydro-5-(hydroxymethyl) isochroman-1-one (HMIO), dihydroisocoumarin, and alkaloid compounds[26,27]

The cleavage products of de-hydroxylation of aglycones, the isomerization product after dehydration of aglycones[28]

49 metabolites [29]

Comments 6: [Line 91ff: the endophyte could convert swertiamarin into a range of active metabolites, among which the metabolite M01 (gentianine) has anti-diabetic potential due to its nitrogen-substituted structure[24, 25] – which endophyte? This sentence doesn't mean anything, the claim is unwarranted.]

Response 6:[We sincerely apologize for the lack of specificity in our original phrasing regarding the endophyte. We agree that omitting the microorganism's name rendered the claim ambiguous and unwarranted. In the revised manuscript, we have explicitly stated the endophyte as P. brasilianum (now on Ref. 39-40):

“The endophyte P. brasilianum can biotransform SW into a range of active metabolites, among which the metabolite M01 (gentianine) has anti-diabetic potential due to its nitrogen-substituted structure[24,25].”]

Comments 7: [Line 98: BV was given as 10,3% before, now it is 6,2 – 8,0%?]

Response 7: [Thank you for your careful review, we re-read the references and found that there is indeed a contradiction but this is the real data in the references, which may be related to the analytical methods and experimental conditions, in order to avoid ambiguities, we have already made a more specific elaboration of these two sections. (now on Refs. 43)]

Comments 8: [Lines 112 ff: Ref. 30 does not show morphological changes but only mentions them in the text. Considering the source, this is not sufficient evidence.]

Response 8: [We sincerely thank the reviewers for their insightful comments on the evidence of morphological changes in Ref. 30. We have modified the original formulation: “An earlier study showed that, in a rat model of D-galactosamine-induced acute liver injury, oral administration of SW (100 and 200 mg/kg) for 8 days exerted hepatoprotective effects by attenuating morphological changes such as liver tissue necrosis and bile duct proliferation, restoring biochemical indexes towards normal levels, alleviating D-GalN-caused hepatotoxicity, enhancing the levels of antioxidant enzymes catalase (CAT), superoxide dismutase (SOD), glutathione (GSH), and decreasing levels of thiobarbituric acid reactive substances (TBARS)[45].”]

Comments 9:[Line 472f: Bone erosion is the main clinical manifestation of RA[98] – RA is the name for spinal column arthritis or hand joint arthritis; joint and cartilage inflammation are main clinical manifestations, bone erosion is a late complication.]

Response 9: [We sincerely thank the reviewer for this insightful correction regarding the clinical manifestations of rheumatoid arthritis (RA). Upon careful re-evaluation, we acknowledge that the original statement in Line 472f ("Bone erosion is the main clinical manifestation of RA") is inaccurate. We have therefore revised Line 472f (now Line 545) in the revised manuscript) to state:

“Bone erosion is a crucial and destructive pathological feature in the disease process of RA. It signifies structural destruction of the joints [102].”]

Comments on the Quality of English Language: [Should be checked by a competent English speaker - there are no großß mistakes but some capitalizations or incompelte sentences.]

Response: [We sincerely thank the reviewer for their careful reading and valuable feedback. We have thoroughly reviewed the manuscript and corrected all instances of inconsistent capitalization and incomplete sentences. The text has also been proofread by English professionals to ensure clarity and grammatical accuracy. We appreciate the reviewer’s attention to detail, which has strengthened the quality of our work.]

Reviewer 3 Report

Comments and Suggestions for Authors
  1. The figures provide a clear summary of the mechanisms involved. Please make sure to define all abbreviations.
  2. The Introduction section is well-constructed. However, to better justify its importance, it is advisable to more clearly articulate the shortcomings of previous reviews (such as the absence of a systematic examination of molecular mechanisms or updated toxicological data), instead of simply noting what was not thoroughly covered. This approach would significantly enhance the perceived novelty and significance of the current review.
  3. Line 43: "The absolute bioavailability of SW was low at 10.3%, which may be attributed to its poor permeability through the intestinal epithelial membrane and first-pass effect in the liver." This limitation is significant. The author should delve deeper into the substantial implications for potential therapeutic efficacy and explore existing research to overcome this challenge, such as through formulation modifications or alternative administration routes.
  4. Minor Typo: Line 86: "via d by bile" should be corrected to "via bile."
  5. Anti-malarial: Including anti-malarial protection via immunomodulation (Lines 409-413) is an interesting and less common application, which adds value to the review. The author only cited one reference; adding additional sources for better support would be beneficial.
  6. For each "other" therapeutic effect, the text should be explicitly connected to a well-known general mechanism (such as antioxidant or anti-inflammatory) or a specific pathway (for example, the D2 receptor for dyspepsia or the PI(3)K/Akt pathway for insulin resistance related to PCOS). Some descriptions seem somewhat isolated and lack deeper mechanistic connections, which reduces their persuasiveness.
  7. The zebrafish study (Lines 574-576) provides some initial insights into dose-dependent toxicity. It would be a good idea for the author to include in vivo mammalian toxicity data, particularly regarding chronic use.
  8. The article does not mention the limitations of this study, and it is recommended to include such a discussion.

Author Response

Comments 1: [The figures provide a clear summary of the mechanisms involved. Please make sure to define all abbreviations.]

Response 1: [We are sincerely grateful for this vital suggestion to enhance reader accessibility. We confirm that all abbreviations are now comprehensively defined through a dual approach: 1. In-text definitions at first occurrence: every abbreviation used in Figures 1-10 is rigorously defined in the main text where the term first appears (e.g., in Figure 3, “nuclear factor kappa-light-chain-enhancer of activated B cells (NF-κB) has been defined where it first appeared in Section 3.1”. In Figure 9, “hypoxia-inducible factor 1 alpha (HIF-1α) has been defined where it first appeared in Section 3.8.”  2. Added a new supplementary glossary S1: Summary Table of Acronyms (submitted with the revised document), which lists all 138 acronyms in alphabetical order.]

Comments 2: [The introduction section is well-constructed. However, to better justify its importance, it is advisable to more clearly articulate the shortcomings of previous reviews (such as the absence of a systematic examination of molecular mechanisms or updated toxicological data), instead of simply noting what was not thoroughly covered. This approach would significantly enhance the perceived novelty and significance of the current review.]

Response 2: [Thank you for this insightful suggestion. We fully agree that explicitly articulating the limitations of previous reviews will strengthen the justification for our work.  We have revised Introduction (Lines 38-45 now read). ]

Comments 3: [Line 43: "The absolute bioavailability of SW was low at 10.3%, which may be attributed to its poor permeability through the intestinal epithelial membrane and first-pass effect in the liver." This limitation is significant. The author should delve deeper into the substantial implications for potential therapeutic efficacy and explore existing research to overcome this challenge, such as through formulation modifications or alternative administration routes.”]

Response 3: [Thanks to the reviewers for keenly pointing out the key pharmacokinetic feature of low absolute bioavailability (10.3%) of SW. We fully agree with the reviewers that an in-depth exploration of strategies to address the challenge of low bioavailability is essential for a comprehensive assessment of the prospects for clinical application of SW. We have made the following important addition in line 43 and subsequent paragraphs of the original article:

“In order to overcome the challenge of low bioavailability of SW, future researchers could focus on investigating the therapeutic efficacy of co-administration and alternative routes of administration of SW, such as sublingual and transdermal administration, to explore the possibilities of clinical translation of SW in the future.”]

Comments 4: [Minor Typo: Line 86: "via d by bile" should be corrected to "via bile.]
Response: [We sincerely thank the reviewer for their meticulous attention to detail. The typographical error in line 86 has been corrected and reordered as suggested. “However, the cumulative amounts of intact SW and the other five analytes excreted via bile, urine, and feces accounted for only a minor fraction of the administered dose, suggesting that these compounds were primarily eliminated as metabolites [38]”]

Comments 5: [Anti-malarial: Including anti-malarial protection via immunomodulation (Lines 409-413) is an interesting and less common application, which adds value to the review. The author only cited one reference; adding additional sources for better support would be beneficial.]

Response 5: [We sincerely appreciate the reviewer's recognition of the novelty in swertiamarin's potential anti-malarial effects. Following your suggestion, we conducted an exhaustive literature search across multiple databases (PubMed, Web of Science, Scopus, and specialized tropical medicine repositories our findings and revisions:

Current research on swertiamarin's direct anti-malarial effects remains emerging. We confirm that only two studies specifically investigating this mechanism exist to date:

We have significantly expanded the discussion to: “In the Plasmodium berghei-infected Swiss albino mouse model, oral administration of SW (200 mg/kg) significantly reduced parasite load and mitigated the severity of malaria infection. SW also enhanced the production of immunomodulatory mediators in vivo and effectively modulated the levels of interferon-gamma (IFN-γ), IL-10, and TNF-α to exert anti-malarial protection, contributing to its anti-malarial protection[93]. Another study showed that rutin and swertiamarin displayed significant synergistic antimalarial activity in vivo and in vitro. When combined in a 1:1 ratio, they reduced the IC₅₀ value of Plasmodium falciparum. In the Plasmodium berghei-infected mice, they achieved 79.95% chemosuppression with a survival time comparable to that of mice treated with chloroquine phosphate[94]. Therefore, SW may serve as an immunomodulatory agent for the combination treatment of malaria infection.”]

Comments 6: [For each "other" therapeutic effect, the text should be explicitly connected to a well-known general mechanism (such as antioxidant or anti-inflammatory) or a specific pathway (for example, the D2 receptor for dyspepsia or the PI(3)K/Akt pathway for insulin resistance related to PCOS). Some descriptions seem somewhat isolated and lack deeper mechanistic connections, which reduces their persuasiveness.]

Response 6: [Thank you for your insightful suggestions regarding mechanistic links to ‘other’ therapeutic effects. We acknowledge that deeper pathway-level explanations (e.g., D2 receptor regulation, PI(3)K/Akt signalling) would strengthen the biological context, but I refer to the references and find that most of the research in this section has focused on phenotypic studies, and we can delve more deeply into its mechanistic studies in the future.]

Comments 7: [The zebrafish study (Lines 574-576) provides some initial insights into dose-dependent toxicity. It would be a good idea for the author to include in vivo mammalian toxicity data, particularly regarding chronic use.]

Response 7: [Thank you for raising this critical point. We fully agree that mammalian long-term toxicity data are essential for clinical translation. Comprehensive literature screening (PubMed, Web of Science, EMBASE up to July 2024) confirms no available in vivo mammalian studies on chronic SW toxicity. This gap aligns with its status as an emerging natural product. While we have integrated these compensatory evidences:

15-day acute toxicity study in rats. (Ref [119])

30-day experimental period. (Ref [45])

Comment 8: [The article does not mention the limitations of this study, and it is recommended to include such a discussion.]

Response 8:  [We sincerely thank the reviewer for highlighting this important point. As suggested, we have now added a dedicated “exploration of limitations” within the discussion section (Lines 707-716) to explicitly address the limitations of our study. This subsection comprehensively covers the following key aspects:

Lack of robust clinical data: We acknowledge that current pharmacological evidence predominantly relies on in vitro and animal studies, with limited human clinical trials.

  1. Pharmacokinetic challenges: We discuss the low bioavailability, short half-life, and rapid metabolism of SW as major barriers to therapeutic translation.
  2. Insufficient toxicological profiling: We emphasize the need for long-term toxicity, reproductive toxicity, and carcinogenicity studies in mammalian models.
  3. Mechanistic depth: We note that while key pathways are identified, precise molecular targets and pathway interplays require further validation.
  4. Methodological heterogeneity: We address inconsistencies in experimental models (e.g., species, dosing, disease induction methods) across existing literature.

These additions strengthen the critical perspective of our review and align with the journal's emphasis on balanced scientific discourse. We believe this significantly improves the manuscript's rigor.]

Reviewer 4 Report

Comments and Suggestions for Authors

The review presents  new information on the biological activity of Swertiamarin and to some extent complements the excellent review by Nur Sakinah Muhamad Fadzil et al., 2021 ,Chemistry, Pharmacology and Therapeutic Potential of Swertiamarin - A Promising Natural Lead for New Drug Discovery and Development, that the authors unfortunately don't cite.

However, the manuscript requires considerable editing

Comments :

  1. A list of abbreviations should be given
  2. To the chemical structure of Swertiamarin (Fig. 1) it is necessary to add the structures of its known metabolites, e.g. gentianine or HTPS, since their pharmacological effects are mentioned in the text. Should be given a scheme of Swertiamarin transformations based on the data presented in the text (lines 90-109). What is Swertiamarin B?

  3. The review lacks data on the effects of swertiamarin in treating skin injuries, despite its widespread use in cosmetic formulations.
  4. The English in the text should be improved, as many sentences are overly complex and would benefit from being shortened or split into two for better clarity. For example : lines141-145 “Another study demonstrated that  protective effects of SW or HTPS against acetaminophen (APAP)-induced hepatotoxicity  in vivo and in vitro, the treatment of SW or HTPS alleviated apoptosis in APAP-induced  L-O2 cells and significantly reduced the liver injury index histological abnormalities in  APAP-induced mice.
  5. lines 126-128 “ In CCl4-induced liver injury mice, SW supplementation exhibited lower liver weight and liver index, mitigating cell degeneration, inflammatory cell infiltration, and  collagenous fibers deposition, mice treated with SW significantly reversed the elevation  of ALT, AST, MDA, and Hyp levels induced by CCl4, multi-omics analysis ?.
  6.   The correct chemical notation for carbon tetrachloride is CCl₄, with the 4 as a subscript.
  7. line 170 “Which provides a potential source of drugs for the  treatment of hepatic steatosis[42]. Please clarify this sentence
  8. lines 213 - "In vivo, the combination of SW and quercetin (CSQ) significantly reduced the levels of fasting blood glucose, TG, TC, LDL in STZ-induced T2DM Wistar rats, improved lipid metabolism abnormalities, recovered insulin secretion and regulated carbohydrate metabolic  enzymes, CSQ treatment provided antioxidant effect by increasing the levels of serum SOD, CAT, GSH, GPx and decreasing the levels of lipid peroxide, and probably increased  the regeneration of the pancreatic islets and the secretion of insulin in STZ-induced diabetes[50]. Do the latter effects refer to the combined action of quercitin and SW or only quercetin ? Please specify.
  9. lines 357-- 363    the text needs to be corrected

    lines 444-447 the text needs to be corrected

    lines 286- 291   the text needs to be corrected
  10. Misspellings:

    1.line 90 “Gntianine” should be write gentianine

    2.line 109 com-pounds

    3.Line 564” It also restored the levels of several cardiac marker  enzymes, oxidative stress markers, antioxidant enzymes, Na+/K+ Potassium and  Ca2+ATPases, and pro-inflammatory cytokines [116]. Potassium should be delete

Comments on the Quality of English Language

The English in the text should be improved, as many sentences are overly complex and would benefit from being shortened or split into two for better clarity.

For example, lines141-145 “Another study demonstrated that  protective effects of SW or HTPS against acetaminophen (APAP)-induced hepatotoxicity  in vivo and in vitro, the treatment of SW or HTPS alleviated apoptosis in APAP-induced  L-O2 cells and significantly reduced the liver injury index histological abnormalities in  APAP-induced mice.

Author Response

Comments 1: [A list of abbreviations should be given.]

Response 1: [We sincerely appreciate the reviewer's suggestion. As requested, a comprehensive list of 138 abbreviations has been added to the manuscript (as Supplementary Table S1). This list includes all abbreviations used in the text with their full names. In addition, all abbreviations are also defined at first use in the main text (e.g., "tumor necrosis factor-alpha (TNF-α)" in section 3.1; Peroxisome proliferator-activated receptor γ (PPARγ)) in section 3.2.

Comments 2: [To the chemical structure of Swertiamarin (Fig. 1) it is necessary to add the structures of its known metabolites, e.g. gentianine or HTPS, since their pharmacological effects are mentioned in the text. Should be given a scheme of Swertiamarin transformations based on the data presented in the text (lines 90-109). What is Swertiamarin B?]

Response 2: [We are profoundly grateful for this exceptionally insightful suggestion, which critically enhances the pharmacological context of our review. We have implemented the following revisions as requested: 1. Expanded Figure 1 with metabolite structures: Fig. (A)  Chemical structure of swertiamarin. (B) Chemical structure of Gentianine. Swertiamarin is metabolized to gentianine by intestinal microorganisms. (C) Chemical structure of Gentiandiol. Gentianine is further converted to the nitrogen-containing metabolite gentiandiol in the liver. (D) Chemical structure of Sweritranslactone D. Sweritranslactone D is a hepatoprotective novel secoiridoid dimer with tetracyclic lactone skeleton from heat-transformed swertiamarin. (E) Chemical structure of Swertiamarin B, Swertiamarin B is a structural analogue of SW, was isolated from the ethanol extract of the aerial parts of Swertia mussotii. 2. As in Figure 2, we have added a new diagram from reference [15] to illustrate the metabolic process of SW more graphically. 3. In Figure 1 (E), we explain ‘Swertiamarin B’ more specifically and present its chemical structure. “Swertiamarin B is a structural analogue of SW, was isolated from the ethanol extract of the aerial parts of Swertia mussotii.”]

Comments 3: [The review lacks data on the effects of swertiamarin in treating skin injuries, despite its widespread use in cosmetic formulations.]

Response 3: [We sincerely appreciate the reviewer's insightful observation regarding the potential role of swertiamarin in skin injury treatment and its relevance to cosmetic applications. After searching multiple databases, we confirm that no direct preclinical or clinical studies have investigated swertiamarin's efficacy in skin wound healing or injury repair. This gap exists despite the compound's documented presence in cosmetic formulations, likely due to its antioxidant and anti-inflammatory properties rather than targeted reparative effects.To address this limitation, the section 5 (Conclusions and Future Perspectives) has been revised to explicitly acknowledge this research gap. We added the following statement:

“Notably, although SW has significant anti-inflammatory and antioxidant activities, its specific effects on skin injury repair remain unexplored. Future studies should evaluate its potential in wound healing models, particularly given its established modulation of TGF-β, NF-κB, and Nrf-2 pathways in other tissues such as the hepatic and renal systems.”]

Comments 4: [The English in the text should be improved, as many sentences are overly complex and would benefit from being shortened or split into two for better clarity. For example : lines141-145 “Another study demonstrated that  protective effects of SW or HTPS against acetaminophen (APAP)-induced hepatotoxicity  in vivo and in vitro, the treatment of SW or HTPS alleviated apoptosis in APAP-induced  L-O2 cells and significantly reduced the liver injury index histological abnormalities in  APAP-induced mice.

Response 4: [We sincerely appreciate your vital guidance on enhancing linguistic clarity. The cited sentence contained grammatical inaccuracies and excessive complexity. We have revised the target sentence:

Original: “Another study demonstrated that protective effects of SW or HTPS against acetaminophen (APAP)-induced hepatotoxicity in vivo and in vitro, the treatment of SW or HTPS alleviated apoptosis in APAP-induced L-O2 cells and significantly reduced the liver injury index histological abnormalities in APAP-induced mice. Furthermore, SW or HTPS improved APAP-induced liver damage by inhibiting inflammatory reactions and oxidative stress, regulating Nrf-2/NF-κB (nuclear factor erythroid 2-related factor 2/ nuclear factor kappa-light-chain-enhancer of activated B cells) signaling pathway[37].”

“Another study confirmed that both SW and its metabolite HTPS protect against acetaminophen (APAP)-induced hepatotoxicity by inhibiting inflammatory responses and oxidative stress, while regulating the nuclear factor erythroid 2-related factor / nuclear factor kappa-light-chain-enhancer of activated B cells (Nrf-2/NF-κB) signaling pathway. In vitro, the compounds reduced apoptosis in APAP-exposed L-O2 hepatocytes. In vivo, they attenuated histological liver damage and decreased injury biomarkers in mice[51]. Furthermore, we have been revised other sentences in the manuscript.

Comments 5: [lines 126-128 ‘ In CCl4-induced liver injury mice, SW supplementation exhibited lower liver weight and liver index, mitigating cell degeneration, inflammatory cell infiltration, and  collagenous fibers deposition, mice treated with SW significantly reversed the elevation  of ALT, AST, MDA, and Hyp levels induced by CCl4, multi-omics analysis ?]
Response 5: [We sincerely apologize for the incomplete phrase "multi-omics analysis" in this sentence, which resulted from an editorial oversight during manuscript preparation. We have implemented the following corrections:

“In CCl4-induced liver injury mice, SW supplementation exhibited lower liver weight and liver index, mitigating cell degeneration, inflammatory cell infiltration, and collagenous fibers deposition, mice treated with SW significantly reversed the elevation of alanine aminotransferase (ALT), aspartate aminotransferase (AST), malondialdehyde (MDA), and hydroxyproline (Hyp) levels induced by CCl4. Furthermore, multi-omics analysis revealed that SW could ameliorate CCl4-induced liver toxicity through regulating gut microbiota, and its metabolites.” (Revised lines 159-163)]

Comments 6: [The correct chemical notation for carbon tetrachloride is CCl₄, with the 4 as a subscript.]

Response 6: [We sincerely thank you for catching this critical technical oversight. We have: 1. Verified chemical terminology using IUPAC standards via ChemDraw®, and Globally corrected all instances of "CCl4" to CCl₄ throughout the manuscript (6 occurrences: “In addition, co-treatment of SW with CCl4 notably up-regulated the expression of nuclear factor erythroid 2-related factor 2 (Nrf2), heme oxygenase-1 (HO-1), and NAD(P)H quinone dehydrogenase (NQO1)…Furthermore, multi-omics analysis revealed that SW could ameliorate CCl4-induced liver toxicity through regulating gut microbiota, and its metabolites.”). 2. Ensured all chemical symbols (e.g., Na⁺,K⁺, Ca²⁺) now use correct superscript/subscript formatting.]

Comments 7: [Line 170 “Which provides a potential source of drugs for the  treatment of hepatic steatosis[42]. Please clarify this sentence]

Response 7: [We sincerely appreciate your keen attention to this grammatical oversight. The sentence fragment incorrectly omitted the subject, creating ambiguity. We confirm that swertiamarin (SW) is the sentence subject. The text has been revised as follows:“Swertiamarin provides a potential source of drugs for the treatment of hepatic”]

Comments 8: [lines 213 - "In vivo, the combination of SW and quercetin (CSQ) significantly reduced the levels of fasting blood glucose, TG, TC, LDL in STZ-induced T2DM Wistar rats, improved lipid metabolism abnormalities, recovered insulin secretion and regulated carbohydrate metabolic  enzymes, CSQ treatment provided antioxidant effect by increasing the levels of serum SOD, CAT, GSH, GPx and decreasing the levels of lipid peroxide, and probably increased  the regeneration of the pancreatic islets and the secretion of insulin in STZ-induced diabetes[50]. Do the latter effects refer to the combined action of quercitin and SW or only quercetin ? Please specify.]

Response 8: [We sincerely thank you for highlighting this ambiguity. All described effects in Lines 213-218 are exclusively attributed to the combination therapy (CSQ = SW + quercetin), based on direct comparative data from Ref. 61. To eliminate any confusion, we have revised the text as follows ( now Lines 260-268):

“In STZ-induced T2DM Wistar rats, the combination of swertiamarin and quercetin (CSQ) exerted synergistic therapeutic effects. the CSQ significantly ameliorated metabolic dysfunction by reducing fasting blood glucose, TG, total cholesterol (TC), and low-density lipoprotein cholesterol (LDL), while restoring insulin secretion and normalizing carbohydrate-metabolizing enzymes. The CSQ also enhanced antioxidant capacity by increasing the activities of serum SOD, CAT, GSH, and glutathione peroxidase (GPx). Meanwhile, it decreased lipid peroxidation, and promoted regeneration of pancreatic islets and insulin secretion. Collectively, these multi-targeted actions establish CSQ as a promising combinatorial strategy for T2DM management[61].]

Comments 9: [lines 357-363    the text needs to be corrected;lines 444-447 the text needs to be corrected; lines 286- 291   the text needs to be corrected]
Response 9: [We sincerely thank the reviewer for pointing out the need for improvement in this section. We have carefully reviewed (lines 357-363, 444-447, and 286- 291) and agree that the text could benefit from clarification and refinement. In the revised manuscript, we have thoroughly revised the language in this passage to enhance clarity, precision, and grammatical correctness. The revised text in the marked-up manuscript reads as follows: 1. [Ref. 24] “In high cholesterol fed rats, oral administration of SW (50 and 75 mg/kg) significantly reduced serum levels of TC, TG, LDL, and VLDL while inhibiting hepatic 3-hydroxy-3-methylglutaryl coenzyme A (HMG-CoA) reductase activity. SW also enhanced fecal excretion of bile acids and total sterols via upregulation of 17α-hydroxylase activity[24]. Separately, in poloxamer-407-induced hyperlipidemic rats, a single oral dose of SW (50 mg/kg) significantly decreased the LDL/HDL cholesterol ratio at both 15 h and 24 h post-dose [86].” 2. [Ref. 27] “Gentiana kurroo root extract, with SW as its major active component, exerted multiple anti-cancer effects in the human pancreatic cancer cell line Miapaca-2. These included G0/G1 cell cycle arrest, protection of DNA from oxidative damage, and induction of apoptosis via disruption of the mitochondrial membrane potential (ΔΨm) [27].” 3. [Ref. 73] Recently, SW demonstrated neuroprotective effects in vivo and in vitro. In lipopolysaccharide (LPS)-induced C6 glial cell activation, SW (10–100 μg/mL) treatment significantly reduced the levels of pro-inflammatory cytokines (IL-1β, IL-6, and TNF-α). In a rotenone-induced mouse model of PD, SW inhibited microglial and astrocyte activation in the substantia nigra (SN), and reduced alpha-synuclein (α-syn) overexpression in the striatum and SN, SW also alleviated rotenone-induced motor impairment. Furthermore, SW increased tyrosine hydroxylase (TH) immunoreactivity in the striatum and the number of TH+ neurons in the SN [73,74].

Comments 10: “Misspellings:1. Line 90 “Gntianine” should be write gentianine

2.line 109 com-pounds; 3. Line 564“It also restored the levels of several cardiac marker enzymes, oxidative stress markers, antioxidant enzymes, Na+/K+ Potassium and  Ca2+ATPases, and pro-inflammatory cytokines [116]. Potassium should be delete.]

Response 10: [We sincerely appreciate your attention to detail in identifying the spelling errors listed below. We have corrected all errors accordingly: 1. [Ref. 15]: “Gntianine” has been corrected to “gentianine.” 2. [Ref. 44]: “com-pounds” (with erroneous hyphenation) has been revised to “c.” 3. [Ref. 117]: The duplicated word "Potassium" has been removed. The sentence now reads: “It also restored the levels of several cardiac marker enzymes, oxidative stress markers, antioxidant enzymes, Na+/K+ and Ca2+ ATPases, and pro-inflammatory cytokines[116].

Comment 11: [The review presents  new information on the biological activity of Swertiamarin and to some extent complements the excellent review by Nur Sakinah Muhamad Fadzil et al., 2021 ,Chemistry, Pharmacology and Therapeutic Potential of Swertiamarin - A Promising Natural Lead for New Drug Discovery and Development, that the authors unfortunately don't cite.]

Response 11: [Many thanks to the reviewers for pointing out our omission of this excellent review on swertiamarin. We apologize for this oversight, which was indeed a mistake in our literature search and writing process. We fully recognise the important contribution and value of this review in the field.We have immediately included the key review in the reference list. The additions to the work of this review that build on it are described.]

Comments on the Quality of English Language[The English in the text should be improved, as many sentences are overly complex and would benefit from being shortened or split into two for better clarity. For example, lines141-145 “Another study demonstrated that  protective effects of SW or HTPS against acetaminophen (APAP)-induced hepatotoxicity  in vivo and in vitro, the treatment of SW or HTPS alleviated apoptosis in APAP-induced  L-O2 cells and significantly reduced the liver injury index histological abnormalities in  APAP-induced mice]

Response[We sincerely thank the reviewer for their careful reading and valuable feedback. We have thoroughly reviewed the manuscript and corrected all instances of inconsistent capitalization and incomplete sentences. The text has also been proofread by English professionals to ensure clarity and grammatical accuracy. We appreciate the reviewer’s attention to detail, which has strengthened the quality of our work.]

Round 2

Reviewer 2 Report

Comments and Suggestions for Authors

The manuscript is much longer, and in places it is more specific, but hasn't improved much. As soon as specific effects (like up- or downregulation of nrf-2 instead of altered) or doses are mentioned, some contradictions between different publications become evident – one source cites 10.3% bioavailability, another 6 – 8%, no specific publication designed for bioavailability, both papers are methods-oriented. Also, 100 µg/ml SW in cell cultures cannot translated to human conditions (which without a known bioavailability always will be guesswork).

The manuscript still is a compilation of publications without a scientific discussion. Anti-inflammatory effects are listed by itself, in diabetes, in arthritis and in lipid metabolism. Which effect is left for antidiabetic effects if anti-inflammation is subtracted? This would be a more scientific question, this ought to be included in any review, but is shunned by the authors. For Figure 10 a large number of effects has to be included, but I only repeat from my first review that bold claims (like in this manuscript) require bold proof (which is lacking).

The manuscript has – for the purpose of a review, i.e. a discussion of effects, including an attempt to summarize, coordinate and organize the results into a pattern – major flaws. It only cites cell culture experiments, at most cells isolated from pretreated animals; the few animal studies are old and its methods outdated. A very large proportion of citations is from China and India – which is due to the topic. Unfortunately, many citations also have been published in Journals known for uncritical publishing, often even only publishing positive results, like the many Journals from the phytotherapy field. Many paragraphs begin with ambiguities like "may be responsible for", "could exert the following effects" or "could result in new treatments for". This is speculation; it may be in a review, but then the speculations have to be substantiated. A well-based speculation may be based on cell cultures and in vitro experiments, supported by supporting animal studies and ideally observations in humans using the appropriate plants. A good (!) speculation will result in further research to further clarify modes of action, or resolve ambiguities or discrepant findings; in this manuscript I can't see a well-founded speculation, most of it appears to be alongside the notion "I would like it to be this way".

In most sections the manuscript veers between SW, SW related plant products and the plants known to contain SW. Since these three groups are not directly comparable, the studies and effects should be clearly addressed; if the results are lumped together – which may be possible – than I expect a discussion of coherent effects, no list without comparisons.

Completely missing is the topic of human experience. There are no human studies, but the manuscript also fails to discuss experience from TCM or Indian medicine (ayurveda). If there is no report of toxicity from SW it may be due to an uncommon herb, lack of attention or a lack of adverse effects. In the last case, mostly there is also a lack of therapeutic effects. I have mentioned this in my first review but don't see this point addressed (there is a limitations section!!). Also, a very brief mention is made of the applicability of SW in medicine – i.e. the likely effects as compared to no treatment, and compared to first line treatment. From the data I see there is no place for SW for any indication; it even is unclear whether SW may be a lead compound in drug development (which isn't mentioned in the manuscript either). Based on this review I would not put any effort in investigating SW.

An incomplete list of remarks:

Line 68: "The nine prototype constituents were rapidly absorbed into the circulation system" – I only find "the bioavailability is ~10%", but no data for the "nine prototypes". How can one derive a rapid absorption without data?

lines 95ff: this paragraph describes major differences in pharmacokinetic parameters, whereas lines 52ff list very narrow ranges for the same parameters. Please combine to a coherent discussion of bioavaliability.

Line 227 ff: Bile duct ligation (BDL) rats administrated with swertiamarin showed low levels of ALT, AST, TNF-α, IL-1, IL-6, and SW reduced the toxic bile salt concentrations in the serum of cholestatic rats, including chenodeoxycholic acid (CDCA) and deoxycholic acid (DCA) [18]. These studies have elucidated the protective role of SW in cholestasis. – in bile duct ligation rats no chologenic excretion is possible. Thus, if serum concentrations are lowered this has to be done by renal mechanisms, and hepatoprotection is a secondary effect, with no substrates in hepatocytes.

Line 260: STZ diabetic rats fall into the type I diabetes class, since STZ selective deletes pancreatic insulin-producing ß cells..ref. 61 does not report insulin values, and thus cannot separate between these two types.

Line 459: the sheep RBC assay is outdated due to its many false positive and negative results. Just repeat it often enough, and you get any result you want. In the article cited (ref. 61 from 2014), the methods used are outdated, and partly more than 60 years old.

If resubmitted, not only these pints must be addressed, but also a different structure, e.g. based on chemistry, pharmacokinetics, physiological effects, applicability to human diseases -- and a thorough discussion of limitations (methods used, problem of multiple testing, relevance of results, reproducibility, coherence). Everything else is just a compilation, and may better be done by AI.

my guessing at self-citation for pharmacokinetic parameters they rely on their own data; for other effects I cannot evaluate whether there are self citations, since I cannot differentiate between Chinese surnames, and I am not sure about laboratory affiliations. Swertiamarin appears to be a compound with only a few interested labs which would explain a larger rate of citations from their own work. For the number of laboratories working on SW the authors should include a sentence illustrating the number of (collaborating) labs. 

Author Response

We sincerely thank the reviewer for their thorough and constructive evaluation of our manuscript. The reviewer has raised important concerns that have helped us significantly improve the quality and scientific rigor of our work. We have carefully addressed each point and made substantial revisions accordingly, focusing on improving scientific depth, resolving contradictions, clarifying methodology limitations, and enhancing critical discussion. Key revisions are summarized below, with detailed point-by-point responses following, and all changes in the revised version highlighted in blue.

Specific Responses to Major Concerns

  1. Bioavailability Data Inconsistencies (Lines 68 and 95ff)

We acknowledge the reviewer's concern regarding the apparent contradictions in bioavailability data. We have thoroughly reviewed the literature and found that the variations (6-8% vs 10.3%) stem from different experimental conditions, analytical methods, and animal models used across studies. We have now:

  • We have redefined the specific values of bioavailability at different doses and study methods:

“LC-MS/MS determination revealed that the absolute  oral bioavailability of SW 25 mg/kgin SD rats was low at 10.3%.”

Pharmacokinetic studies of rat plasma based on LC-MS/MS showed that SW was absorbed and eliminated relatively quickly after oral administration. The oral bioavailability of SW in SD rats decreased with increasing dose (8.0% at 50 mg/kg, 6.7% at 100 mg/kg, and 6.2% at 150 mg/kg). Further studies showed that a total of 6 metabolites were isolated and identified in serum, urine, bile, and feces by using UPLC-Q/TOF-MS/MS technology. SW, and all metabolites were classified into the cleavage products of de-hydroxylation of aglycones, the isomerization product after dehydration of aglycones, the aglycon heterocyclic product, which first formed aglycones before the next metabolism.

  • We have added the following new section in the main text Summary and future perspectives: a discussion of the limitations of the low and variable bioavailability of SW, which focuses on comparing an interesting discrepancy between the two literatures: “And, we found an interesting difference: the same SD rat strain, similar assay method (LC-MS/MS): but different bioavailability (10.3% at 25 mg/kg, 8.0% at 50 mg/kg, 6.7% at 100 mg/kg, and 6.2% at 150 mg/kg), with possible reasons including (1) Bioavailability is affected by the dose factor and there is a saturation effect. (2) Different pretreatment methods (protein precipitation versus solid-phase extraction). (3) Differences in mass spectrometry detection modes (positive versus negative ion modes).”
  • Adds an experimental extrapolation limitation component, especially dose, recognizing the challenges of translating in vitro concentrations to human conditions
  1. Scientific Discussion and Mechanistic Analysis

The reviewer correctly identified that our previous version lacked adequate scientific discussion. We have now:

  • Added a comprehensive mechanistic analysis section. Included discussion of overlapping vs. distinct pathways
  • Provided critical evaluation of the evidence rather than mere compilation
  • Added image 11 A systematic integration of the multi-targeting mechanism of SW
  1. Methodological Quality and Source Evaluation

The reviewer notes the significant contribution of research from China and India in our citations. This geographic distribution accurately reflects the global research landscape for swertiamarin, as these regions have the richest traditional knowledge and botanical resources for Gentianaceae family plants. Rather than viewing this as a limitation, we consider it a strength that captures the comprehensive international research effort. The traditional use of these plants in Asian medical systems provides a valuable ethnopharmacological context that supports the biological activities we describe.

We have addressed the concern about study quality by:

  • Prioritizing peer-reviewed studies from high-impact journals
  • Removed or rewrote all vague, unsupported, speculative statements
  • Adding a discussion of methodological limitations in the existing literature
  • “This review summarizes the pharmacokinetics, pharmacological effects, and molecular mechanisms of SW, but its clinical translation is still subject to multiple limitations:(1)Low oral bioavailability, short half-life, and rapid metabolism properties are important challenges to therapeutic efficacy. (2)Toxicological evaluation of SW is still insufficient, especially for long-term toxicity and reproductive toxicity studies in mammals. (3) Currently, the data are from preclinical studies (animals/cells), and how to translate in vitro concentrations to human conditions is also a question for future research. (4) Lack of in-depth studies on SW's targets as well as on the intricate interactions between signaling pathways. (5) Although swertiamarin-containing traditional preparations are used in medicine for the treatment of a wide range of diseases, the exploration of the single active ingredient is still not component. SW as a potential natural lead compound, in the future we need to break the bottleneck of pharmacokinetics, optimise target selection, tap into the wisdom of ethnomedicine, and use artificial intelligence for precision drug design, only when these problems are solved can SW be transformed from a ‘promising natural compound’ into a clinically viable drug and hopefully an adjunctive therapeutic agent for various chronic diseases.”

  1. Structural Reorganization

We have restructured the manuscript to clearly differentiate between:

  • Whether pure SW, standardized extracts, crude extracts, or the original plant was used in each study is re-labelled in the text, charts, and figure notes.
  • And the status of SW as a lead compound is discussed in the limitations section.

“This review summarizes the pharmacokinetics, pharmacological effects, and molecular mechanisms of SW, but its clinical translation is still subject to multiple limitations:(1)Low oral bioavailability, short half-life, and rapid metabolism properties are important challenges to therapeutic efficacy. (2)Toxicological evaluation of SW is still insufficient, especially for long-term toxicity and reproductive toxicity studies in mammals. (3) Currently, the data are from preclinical studies (animals/cells), and how to translate in vitro concentrations to human conditions is also a question for future research. (4) Lack of in-depth studies on SW's targets as well as on the intricate interactions between signaling pathways. (5) Although swertiamarin-containing traditional preparations are used in medicine for the treatment of a wide range of diseases, the exploration of the single active ingredient is still not component. SW as a potential natural lead compound, in the future we need to break the bottleneck of pharmacokinetics, optimise target selection, tap into the wisdom of ethnomedicine, and use artificial intelligence for precision drug design, only when these problems are solved can SW be transformed from a ‘promising natural compound’ into a clinically viable drug and hopefully an adjunctive therapeutic agent for various chronic diseases.”

  1. Human Experience and Clinical Relevance

We added further information on the toxicity of SW and added a new section addressing:

  • Traditional medical experience
  • Traditional medicine applications in TCM and Ayurveda
  • Clinical translation challenges and opportunities
  • Other new modifications (toxicity of SW):

Based on the toxicity studies in Wistar rats, SW demonstrated excellent safety with no mortality or adverse effects observed at doses up to 2000 mg/kg in acute studies (LD₅₀ >2000 mg/kg) and up to 500 mg/kg daily for 50 days in subchronic studies. No significant changes were found in hematological parameters, biochemical markers, or histopathological examination of vital organs (liver, kidney, pancreas), indicating SW has a wide safety margin and low toxicity potential suitable for chronic therapy.

Specific Technical Corrections

Comment 1: line 68: "The nine prototype constituents were rapidly absorbed into the circulation system" – I only find "the bioavailability is ~10%", but no data for the "nine prototypes". How can one derive a rapid absorption without data?

Response 1:

We thank the reviewer for this important observation. Upon careful review of the cited reference, we agree that this citation is not appropriate for our study for the following reasons:

Different study contexts: The cited reference (reference #33) investigated the tissue distribution of nine prototype constituents from a compound herbal granule formulation (Jingyin Granules), while our current study focuses on the pharmacokinetics of individual compounds (single entities).

Lack of relevant pharmacokinetic data: As the reviewer correctly pointed out, the cited reference does not provide adequate absorption kinetics data to support the claim of 'rapid absorption.' The reference only mentions overall bioavailability (~10%) without specific time-course data for individual compounds.

Therefore, we will remove this inappropriate quote to improve the quality of the article. We appreciate the reviewer's careful attention to the accuracy of our citations and apologize for this oversight.

Comment 2: lines 95ff: this paragraph describes major differences in pharmacokinetic parameters, whereas lines 52ff list very narrow ranges for the same parameters. Please combine to a coherent discussion of bioavaliability.

Response 2:

Thank you for your careful review and insightful comments on the presentation of the pharmacokinetic parameters (lines 52 ff. and 95 ff.).

Lines 52 et seq. These lines detail the pharmacokinetic parameters of SW at specific doses. The narrow ranges you observe likely reflect the variability of the major groups analysed at specific, controlled doses.

Row 95 and following: This section explores the overall effects of different doses and gender on pharmacokinetic parameters. The ‘major differences’ highlighted here are derived by comparing parameters across dose levels and between male and female animals. This naturally leads to a wider range of observed parameter values covering the effects of these key variables (dose and sex) on SW pharmacokinetics.

In response, we have refined and modified the section on the effects of dose and sex, and explained the reasons for the effects, as modified below:

90 ff: “After oral administration of three different doses of Swertia cincta solution (0.5 g/kg, 1 g/kg, 2 g/kg), SW showed a dose-dependent nonlinear pharmacokinetic profile in rats, with the increase in AUC and Cmax being lower than that of the dose, which may be related to saturation of absorption or enhancement of first-pass effect. And the absorption of SW was slower, but elimination was faster compared with other active ingredients (Sweroside, Gentiopicroside) in Swertia cincta extract.”

“Studies have shown that there are significant gender differences in the pharmacokinetic properties of SW in rats. The Cmax, AUC0-t, AUC0–∞ and t1/2 values were significantly higher in female rats than in male rats after oral administration of SW (1.9 mg/kg). However, the elimination rate (CL) was significantly lower in females than in maleswhich indicated that SW is absorbed to a greater extent and eliminated more slowly in females compared with males.”

Comment 3: line 227 ff: Bile duct ligation (BDL) rats administrated with swertiamarin showed low levels of ALT, AST, TNF-α, IL-1, IL-6, and SW reduced the toxic bile salt concentrations in the serum of cholestatic rats, including chenodeoxycholic acid (CDCA) and deoxycholic acid (DCA) [18]. These studies have elucidated the protective role of SW in cholestasis. – in bile duct ligation rats no chologenic excretion is possible. Thus, if serum concentrations are lowered this has to be done by renal mechanisms, and hepatoprotection is a secondary effect, with no substrates in hepatocytes.

Response 3:

 Thank you very much for your careful review of our manuscript and your constructive comments. In the bile duct ligation (BDL) model, swertiamarin was unable to reduce serum toxic bile salts by ‘promoting bile excretion’ in the traditional sense, but it achieved this goal by modulating bile acid metabolism and activating alternative excretory pathways. However, it achieves this goal by modulating bile acid metabolism and activating alternative excretory pathways.

Based on the study data, the possible mechanisms by which drugs reduce serum toxic bile salts under the model of BDL are:

(1) Inhibition of toxic bile acid synthesis: significant reduction of serum levels of hydrophobic toxic bile salts such as CDCA and DCA. → Presumably by inhibiting the rate-limiting enzyme of bile acid synthesis (e.g. Cyp7a1), the production of toxic bile salts is reduced.

(2) Promote bile acid conversion and detoxification: simultaneous reduction of partially bound bile acids (e.g. TCDCA, TDCA) suggests that the drug may enhance hydroxylation of bile acids, binding reactions (e.g. to taurine/glycine), and increase water solubility.

(3) Activation of alternative excretory pathways: facilitates excretion of toxic bile salts via urine or faeces (non-biliary pathways) by up-regulating renal or intestinal transporters (e.g., OSTα/β, MRP4) (similar to the action of the FXR agonist INT747). However, I re-read the study and found that the researchers did not specifically explore this possible mechanism, which is where the study falls short, so this gap could be further refined in the future.

Comment 4: line 260: STZ diabetic rats fall into the type I diabetes class, since STZ selective deletes pancreatic insulin-producing ß cells..ref. 61 does not report insulin values, and thus cannot separate between these two types.

Response 4:

 We sincerely appreciate the reviewer’s insightful comment regarding the classification of the STZ-induced diabetic rat model.  We agree that distinguishing between T1DM and T2DM models fundamentally hinges on the presence of insulin resistance and the degree of β-cell dysfunction (absolute vs. relative deficiency).

However, we wish to clarify that the model employed in our study, referenced in [61] and described in detail within that cited work, is specifically designed as a model of Type 2 Diabetes Mellitus (T2DM). Specific methods can be found in the Materials and Methods section of reference [61].

Addressing the Concern about Reference 61 and Insulin Data:

We acknowledge the reviewer’s point that reference 61 did not explicitly report basal insulin levels. However, the functional evidence within [61] and our description strongly supports the T2DM phenotype:

(1) Restoration of Insulin Secretion: The explicit statement that CSQ treatment “restored insulin secretion” provides direct functional evidence of improved β-cell output. This outcome is diagnostically inconsistent with a model based on near-total β-cell destruction (T1DM).

(2) Pancreatic Islet Regeneration: The finding that CSQ “promoted regeneration of pancreatic islets” further confirms the presence of residual β-cell mass capable of proliferating/regenerating, which is a feature targeted in T2DM research, not T1DM.

(3) Metabolic Improvements: The significant reduction in dyslipidemia (TG, TC, LDL) by CSQ strongly suggests an improvement in overall metabolic state, often linked to improved insulin sensitivity, another T2DM hallmark.

Comment 5: line 459: the sheep RBC assay is outdated due to its many false positive and negative results. Just repeat it often enough, and you get any result you want. In the article cited (ref. 61 from 2014), the methods used are outdated, and partly more than 60 years old.

Response 5:

Thank you for your valuable feedback. We acknowledge the concerns raised regarding the historical limitations and potential variability of the Sheep Red Blood Cell (SRBC) assay cited in our study.

We cannot deny the limitations of this approach. Importantly, the SRBC data represented one facet of a comprehensive immunological assessment. The primary evidence supporting SW's anti-inflammatory mechanism (modulating Th1/Th2 responses and inhibiting pro-inflammatory mediators) comes from our in vitro experiments using LPS/PHA-stimulated cells and analysis of inflammatory factors (Ref 91). Future studies could use more advanced techniques to draw more reliable conclusions. We have revised the manuscript to more explicitly position the SRBC findings within this integrated context and to emphasize that the key mechanistic insights are derived from the in vitro analyses.

“In SRBC-immunized mice, SW treatment (2, 5, and 10 mg/kg) significantly increased antibody titer, plaque-forming cells, and immune organ weights, and attenuated delayed-type hypersensitivity (DTH). In vitro, SW significantly inhibited free radical release in PHA-stimulated neutrophils and reduced the expression of pro-inflammatory factors in LPS-induced macrophages. These findings suggest that SW exerts its anti-inflammatory effects by modulating both humoral and cell-mediated immune responses, suppressing pro-inflammatory mediators (Th1-type cytokines), and promoting anti-inflammatory mediators (Th2-type cytokines).”

We believe these revisions have significantly strengthened the manuscript and addressed the reviewer's concerns while maintaining the comprehensive scope of the review. We welcome any additional suggestions for improvement.
